# GPT CAN SOLVE MATHEMATICAL PROBLEMS WITHOUT A CALCULATOR

## ABSTRACT

Previous studies have typically assumed that large language models are unable to accurately perform arithmetic operations, particularly multiplication of >8 digits, and operations involving decimals and fractions, without the use of calculator tools. This paper aims to challenge this misconception. With sufficient training data, a 2 billion-parameter language model can accurately perform multi-digit arithmetic operations with almost 100% accuracy without data leakage, significantly surpassing GPT-4 (whose multi-digit multiplication accuracy is only 4.3%). We also demonstrate that our MathGLM, fine-tuned from GLM-10B on a dataset with additional multi-step arithmetic operations and math problems described in text, achieves similar performance to GPT-4 on a 5,000-samples Chinese math problem test set.

## 1 INTRODUCTION

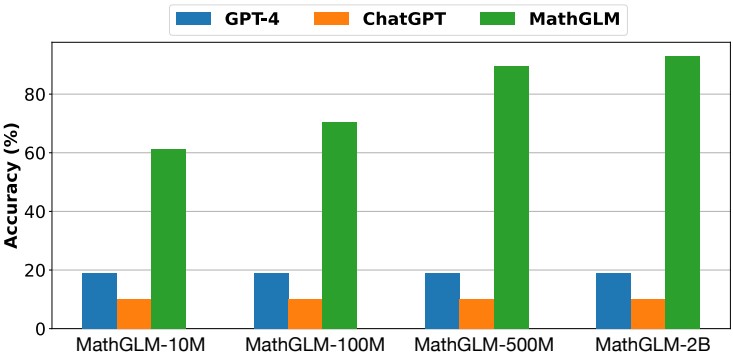

Figure 1: Accuracy scores across various LLMs like GPT-4 and ChatGPT, as well as a series of MathGLM models on the generated test dataset for the arithmetic tasks. Among the different model scales, MathGLM consistently achieves superior performance.

Large language models (LLMs) have demonstrated remarkable ability in handling a variety of downstream tasks in the NLP domain (Brown et al., 2020; Chowdhery et al., 2022; Zeng et al., 2022; Thoppilan et al., 2022; Zhang et al., 2022b; Scao et al., 2022). Pioneering models, such as GPT-4 (OpenAI, 2023) and ChatGPT (OpenAI), have been trained on massive amounts of text data, enabling them to generate coherent and contextually relevant responses. Their ability to understand and generate text makes them highly versatile for various NLP tasks. Moreover, LLMs have been leveraged for other assignments, involving areas of mathematics (Cobbe et al., 2021; Lewkowycz et al., 2022) and science (Taylor et al., 2022). Nevertheless, despite the impressive capabilities across diverse NLP tasks, GPT-4 might not exhibit the same level of proficiency in mathematical reasoning, including arithmetic tasks and Chinese math word problems.

In the context of arithmetic tasks, a prevailing assumption is that LLMs struggle with accurately executing complex arithmetic operations, especially pronounced in cases involving multiplication of numbers exceeding 8 digits, and operations entailing decimals and fractions. To eliminate these misconceptions, we embark on an investigation to assess the arithmetic ability of LLMs. Specifically,

we focus on the capability of LLMs in performing complex arithmetic operations. As a result, we propose MathGLM, a powerful model meticulously crafted to impeccably execute an extensive spectrum of complex arithmetic operations, achieving the best performance compared to leading LLMs such as GPT-4 (See Figure 1). These operations contain singular actions like addition, subtraction, multiplication, division, and exponentiation, as well as the mixing of these operations employing brackets. When these operations are performed individually, without being combined with any other operation, we refer to them as "1-atomic operation". Importantly, MathGLM has the capability to adeptly tackle arithmetic operations that involve a variety of numerical forms, including integers, decimals, fractions, percentages, and even negative numbers.

To attain the remarkable performance exhibited by MathGLM in arithmetic tasks, we utilize a step-by-step strategy to construct an arithmetic dataset that serves as the foundation for MathGLM's pre-training. This dataset is designed to encompass a wide spectrum of arithmetic operations, spanning from straightforward 1-atomic operation to more complex 9-atomic operations. By adopting this step-by-step strategy, MathGLM learns to handle both simple and intricate arithmetic expressions, which empowers it to accurately perform calculations even for operations involving multiplication of numbers greater than 8 digits, and those with decimals and fractions. Moreover, we incorporate the concept of curriculum learning to further augment the capabilities of MathGLM. By gradually increasing the complexity of the arithmetic expressions, MathGLM progressively enhances its capacity to tackle operations involving numbers spanning up to 12 digits. This stands in contrast to the common assumption that large language models struggle with such complex arithmetic tasks. The results demonstrate that MathGLM's arithmetic performance surpasses even the most robust LLMs like GPT-4. Specifically, MathGLM achieves an impressive accuracy of 93.03% on the test dataset containing complex mixed operations. In contrast, GPT-4 only manages a meager 18.84% accuracy on the same dataset.

For math word problems, the Ape210K dataset (Zhao et al., 2020) serves as a comprehensive source of mathematical challenges, drawing from diverse math word problems across the Internet. This dataset serves as a valuable resource for training MathGLM, offering a broad spectrum of problem types for learning. However, a notable characteristic of the original dataset lies in its directly calculated answers. This straightforward answer presentation might lead to a potential drawback, that is MathGLM can potentially miss the underlying calculation rules and patterns embedded within the calculation processes.

To overcome this potential limitation and bolster MathGLM's proficiency in solving math word problems, we leverage the step-by-step strategy to reconstruct the Ape210K dataset. By decomposing the complex arithmetic calculation process into a sequence of sequential steps, MathGLM is empowered to accurately generate answer for math word problems and significantly enhance the answer accuracy in comparison to the original one. For instance, MathGLM achieves an impressive absolute gain of 42.29% in answer accuracy as compared to fine-tuning on the original dataset. By fine-tuning from the GLM-10B, MathGLM's performance closely aligns with that of GPT-4 when evaluated on a math word problems dataset comprising 5,000 test cases. This step-by-step strategy provides MathGLM with a profound understanding of the complex calculation process inherent in math word problems, enabling MathGLM to grasp the underlying calculation rules and obtain more accurate answers.

Overall, MathGLM excels in both arithmetic tasks and math word problems by leveraging the step-by-step strategy. Our comprehensive experiments and detailed analysis demonstrate the effectiveness of MathGLM's mathematical reasoning compared to GPT-4. These results significantly challenge the common misconception that LLMs struggle with complex arithmetic tasks, thus unveiling their remarkable potential to excel in the realm of mathematical reasoning tasks.

## 2 METHOD

To investigate the efficacy of LLMs in mathematical reasoning, we propose the MathGLM model that designed with the specific goal of enhancing the performance of LLMs in mathematical reasoning. **Firstly**, MathGLM focuses on enhancing its proficiency in accurately executing a comprehensive range of arithmetic tasks. It accomplishes this by integrating a step-by-step strategy into its architecture. Instead of straightforwardly calculating the answers to complex arithmetic expressions, MathGLM employs this strategy to meticulously generate answers step by step. **Secondly**, MathGLM leverages the step-by-step strategy to fine-tune a series of GLM models on specific Chinese math-

emratical problems. By leveraging this strategy, MathGLM enhances its ability to handle complex mathematical problem-solving tasks.

## 2.1 LEARNING ON ARITHMETIC TASKS

Arithmetic tasks can be broadly divided into basic arithmetic operations and complex mixing operations. Basic arithmetic operations encompass fundamental mathematical tasks that revolve around conducting simple calculations involving two numbers. On the other hand, arithmetic tasks also encompass the domain of complex mixing operations, which necessitate the skill to manage a combination of diverse arithmetic operations and numerical formats. A comprehensive category of the learning tasks encompassed by MathGLM is summarized in Table 1.

Table 1: Summary and symbolic expression of arithmetic tasks. In symbolic expression, we represent a decimal with n-digit integer part and m-digit decimal part as nD.mD. For mixed computing, we only show a simple mixed symbolic expression.

| Task | Integer | Decimal | Fraction | Percentage | Negative Numbers |
|------|---------|---------|----------|------------|------------------|
| Addition | nD+nD | nD.mD+nD.mD | (nD/mD)+(nD/mD) | nD%+nD% | -nD+-nD |
| Subtraction | nD-nD | nD.mD-nD.mD | (nD/mD)-(nD/mD) | nD%-nD% | -nD–nD |
| Multiplication | nD*nD | nD.mD*nD.mD | (nD/mD)*(nD/mD) | nD%*nD% | -nD*-nD |
| Division | nD/nD | nD.mD/nD.mD | (nD/mD)/(nD/mD) | nD%/nD% | -nD/-nD |
| Exponentiation | nD^nD | - | - | - | -nD^-nD |
| Mixed Computing | | | [(nD±nD.mD)*nD%]/-nD | | |

To augment the arithmetic ability of MathGLM, we adopt a decoder-only architecture based on Transformer (Vaswani et al., 2017) and train it from scratch on our generated arithmetic dataset using an autoregressive objective.

**Arithmetic Dataset.** The arithmetic dataset employed for pre-training is meticulously designed to encompass a comprehensive range of arithmetic tasks. This dataset is thoughtfully designed to incorporate a variety of operations, including addition, subtraction, multiplication, division, and exponentiation. Additionally, it encompasses diverse numerical formats such as integers, decimals, percents, fractions, and negative numbers. This comprehensive dataset is created in various sizes, ranging from 1 million to 50 million records. Within each of these datasets, individual arithmetic expressions range from 1 to 9 atomic operations, including mathematical functions such as addition (+), subtraction (-), multiplication ($\times$), division (/), and exponentiation ($\hat{}$). To align with human calculation habits, a step-by-step strategy is employed in the construction of the arithmetic datasets. Instead of directly computing the final answer to each complex arithmetic expression, the strategy breaks down the complex expression into a sequence of simpler steps, progressively generating answers step by step. This strategy mirrors the process human typically follow when solving complex arithmetic tasks. By training on such dataset, MathGLM achieves outstanding arithmetic performance since it learns the underlying calculation rules from the detailed calculation process. Figure 2 provides some training examples drawn from the arithmetic dataset, illustrating the diversity of arithmetic tasks and the step-by-step strategy incorporated in the dataset. For a more in-depth exploration of the specifics of the generated datasets, the details can be found in Appendix A.1.

**Models and Training Procedure.** Our training efforts encompass 4 distinct types of models, each characterized by different parameter sizes. The largest model is endowed with 2B parameters, making it the most powerful in terms of capacity. Following that, we train the second model with 500M parameters, the third model with 100M parameters and the smallest model with 10M parameters. Notably, despite the discrepancies in parameter sizes, all models are trained using the same dataset scale consisting of 50 million training records. A comprehensive overview of the MathGLM, including its various training parameters and tokenization are presented in Appendix B.1.

For training procedure, we employ the fundamental principle of curriculum learning to effectively train the MathGLM. The training procedure of MathGLM is initiated using an arithmetic dataset containing numbers within a range of 5 digits. Following this initial phase, where MathGLM

| Basic arithmetic operations | Complex mixing operations |
|---|---|
| 1+8/1*10+2=1+8*10+2=1+80+2=81+2=83 | 7826+855+4919/1050*1362-3673*7531/6726+5633=7826+855+4.6847619047619045*1362-3673*7531/6726+5633=7826+855+6380.645714285713-3673*7531/6726+5633=7826+855+6380.645714285713-27661363/6726+5633=7826+855+6380.645714285713-4112.602289622361+5633=8681+6380.645714285713-4112.602289622361+5633=15061.645714285714-4112.602289622361+5633=10949.043424663352+5633=16582.043424663352 |
| 7/5/8-3=1.4/8-3=0.175-3=-2.825 | |
| 5+5+8/1*2=5+5+8*2=5+5+16=10+16=26 | |
| 5/9=0.5555555555555556 | |
| 6*5*4=30*4=120 | 674+2939*2987*9430+6994/883-1642/521+2051=674+8778793*9430+6994/883-1642/521+2051=674+82784017990+6994/883-1642/521+2051=674+82784017990+7.920724801812004-1642/521+2051=674+82784017990+7.920724801812004-3.1516314779270633+2051=82784018664+7.920724801812004-3.1516314779270633+2051=82784018671.92073-3.1516314779270633+2051=82784018668.7691+2051=82784020719.7691 |
| 5/8=0.625 | |
| 7*10+6=70+6=76 | |
| 4+9=13 | |
| 2+9-8=11-8=3 | [(-4453+9698.9284)*-4992.0]*3575/3238+-4722.991=[5245.928400000001*-4992.0]*3575/3238+-4722.991=(-5245.928400000001*4992.0)*3575/3238+-4722.991=(-26187674.572800003)*3575/3238+-4722.991=-26187674.572800003*3575/3238-4722.991=-26187674.572800003*3575/3238-4722.991=-93620936597.76001/3238-4722.991=-28913198.455145154-4722.991=-28917921.446145155 |
| 2-1=1 | |
| 3/2/7=1.5/7=0.21428571428571427 | |
| 7*6*4=42*4=168 | |
| 5*6/10-6*9=30/10-6*9=3-6*9=3-54=-51 | (-8174.1-4561%)/-727.36226-8943=(-8174.1-45.61)/-727.36226-8943=-8219.71/-727.36226-8943=8219.71/727.36226-8943=11.300710047837788-8943=-8931.699289952163 |
| 2+8/1=2+8=10 | |
| 6/10=0.6 | 8689%*-5814*190-6470%/[-5900-(3540%/5945)]=86.89*-5814*190-64.7/[-5900-(35.4/5945)]=86.89*-5814*190-64.7/[-5900-0.005954583683767872]=86.89*-5814*190-64.7/-5900.005954583684=-86.89*5814*190+64.7/5900.005954583684=-505178.46*190+64.7/5900.005954583684=-95983907.4+64.7/5900.005954583684=-95983907.4+0.010966090627372149=-95983907.38903391 |
| 4+7-1+4-10=11-1+4-10=10+4-10=14-10=4 | |
| 1/5+9=0.2+9=9.2 | |
| 9*9=81 | |

Figure 2: Some examples of the arithmetic training dataset of MathGLM.

attains stable training convergence and demonstrates satisfactory performance on the test dataset, we introduce curriculum learning to enhance its capabilities. Specifically, we augment the training data with a new dataset comprising 50,000 records, which encompass numbers spanning from 5 to 12 digits. By incorporating these more challenging examples, MathGLM is encouraged to decipher the rules associated with arithmetic operations involving large numbers. Such training strategy allows MathGLM initially tackles simpler examples, progressively advancing towards more complex challenges. More importantly, such approach empowers MathGLM to improve its ability by learning from relatively smaller examples, emphasizing the efficiency of MathGLM to handle increasingly intricate tasks or data patterns.

## 2.2 LEARNING ON MATH WORD PROBLEMS

Alongside our focus on arithmetic tasks, we train (fine-tune) a series of Transformer-based language models, named General Language Model (GLM) (Du et al., 2021; Zeng et al., 2022) and their chat versions to solve math word problems (MWP). Our training leverages the publicly available Chinese Ape210K dataset, which serves as a valuable resource for training language models on math word problem-solving tasks. This dataset consists of a vast collection of 210,000 Chinese math problems at the primary school level, with each problem's answer calculated directly.

**Dataset.** To enhance the performance of MathGLM on MWP, we utilize a step-by-step strategy to reconstruct the Ape210K dataset where the answer of each math problem is calculated step by step. Figure 3 demonstrate the contrast between the original Ape210K dataset and our reconstructed version. The newly reconstructed dataset encourages MathGLM to acquire an in-depth understanding of the underlying calculation rules inherent in solving math word problems. Through this step-wise process, MathGLM becomes adept at deriving a final, accurate answer for each problem, emphasizing its ability to harness the complexities of mathematical reasoning. The validation dataset used to evaluate the effectiveness of MathGLM can be found in Appendix A.2.

**Backbone Models.** We adopt different variations of the GLM as the backbone to train the MathGLM, including GLM-large with 335M parameters, GLM-6B, GLM2-6B, and GLM-10B. Besides, we train the MathGLM using the ChatGLM-6B and ChatGLM2-6B backbones. These backbone models bestow the MathGLM with a basic language understanding skills, enabling it to effectively comprehend linguistic information contained within math word problems. The details of backbone models are presented in Appendix B.2.

| The original Ape210K dataset | The reconstructed Ape210K dataset |
|---|---|
| {"question": "小王要将150千克含药量20%的农药稀释成含药量5%的药水．需要加水多少千克？", "answer": "x=150*20%/5%−150=450"} | {"question": "小王要将150千克含药量20%的农药稀释成含药量5%的药水．需要加水多少千克？", "answer": "x=150*20%/5%−150=150*0.2/0.05−150=30/0.05−150=600−150=450"} |
| {"question": "Xiao Wang wants to dilute 150 kilograms of pesticides with a pesticide content of 20% into a potion with a pesticide content of 5%. How many kilograms of water need to be added? ", "answer": "x=150*20%/5%-150=450"} | {"question": "Xiao Wang wants to dilute 150 kilograms of pesticides with a pesticide content of 20% into a potion with a pesticide content of 5%. How many kilograms of water need to be added? ", "answer": "x=150*20%/5%-150=150*0.2/0.05-150=30/0.05-150=600-150=450"} |
| {"question": "一个圆形花坛的半径是4米，现在要扩建花坛，将半径增加1米，这时花坛的占地面积增加了多少米**2．", "answer": "x=(3.14*(4+1)**2)−(3.14*4**2)=28.26"} | {"question": "一个圆形花坛的半径是4米，现在要扩建花坛，将半径增加1米，这时花坛的占地面积增加了多少米**2．", "answer": "x=(3.14*(4+1)**2)−(3.14*4**2)=(3.14*5**2)−(3.14*4**2)=(3.14*25)−(3.14*4**2)=78.5−(3.14*4**2)=78.5−(3.14*16)=78.5−50.24=28.26"} |
| {"question":"The radius of a circular flower bed is 4 meters. Now we need to expand the flower bed and increase the radius by 1 meter. How many meters will the area of the flower bed increase at this time** 2.", "answer": "x=(3.14*(4+1)**2)-(3.14*4**2)=28.26"} | {"question":"The radius of a circular flower bed is 4 meters. Now we need to expand the flower bed and increase the radius by 1 meter. How many meters will the area of the flower bed increase at this time** 2.", "answer": "x=(3.14*(4+1)**2)-(3.14*4**2)=(3.14*5**2)-(3.14*4**2)=(3.14*25)-(3.14*4**2)=78.5-(3.14*4**2)=78.5-(3.14*16)=78.5-50.24=28.26"} |
| {"question": "一个三角形的面积是32cm**2，底是8cm，高是多少cm．", "answer": "x=32*2/8=8"} | {"question": "一个三角形的面积是32cm**2，底是8cm，高是多少cm．", "answer": "x=32*2/8=64/8=8"} |
| {"question": "The area of a triangle is 32cm**2, the base is 8cm, and the height is how many cm.", "answer": "x=32*2/8=8"} | {"question": "The area of a triangle is 32cm**2, the base is 8cm, and the height is how many cm.", "answer": "x=32*2/8=64/8=8"} |

Figure 3: Comparison between the original Ape210k dataset and the reconstructed version. A step-by-step strategy is employed to reconstruct the solutions for each mathematical problem.

**Training Strategy.** To achieve better performance, we employ two training strategies for MathGLM. The first is to fine-tune the GLM backbone models on a solitary mathematical dataset. This process allows the MathGLM to specialize in understanding and solving math word problems by learning from the mathematical dataset's unique characteristics. However, such strategy damages the generic ability of the MathGLM. To circumvent this limitation, a second strategy is to continue training the GLM backbone models on a hybrid dataset that combines both mathmatics and text content.

## 3 EXPERIMENTS

The overarching objective of MathGLM revolves around demonstrating the ability of language models in the domain of mathematical reasoning. To validate this, we design two distinct types of experiments, encompassing arithmetic tasks and math word problems. Here, we utilize *Accuracy* and *RE* to measure the ability of MathGLM on arithmetic tasks. The details of evaluation metrics is presented in Appendix C.

### 3.1 LEARNING ON ARITHMETIC

**Overall Results.** We contrast the performance between MathGLM with those of leading LLMs such as GPT-4 and ChatGPT. As presented in Table 2, MathGLM consistently outperforms all other models, indicating its superior performance in tackling arithmetic tasks. Even when we consider a more small model variant, namely MathGLM-10M with a mere 10 million parameters, the results reveal a surprising phenomenon. Despite its compact parameter size, MathGLM-10M outperforms GPT-4 and ChatGPT across an array of comprehensive arithmetic tasks. This astonishing results show the effectiveness of MathGLM, which involves decomposing complex arithmetic expressions into individual steps, granting it the capacity to discern and comprehend the subtleties within arithmetic tasks. It effectively learns the underlying rules and principles of arithmetic operations, enabling it to generate accurate and precise solutions. Furthermore, when comparing MathGLM across different parameter scales, we observe that the MathGLM's arithmetic performance is directly correlated with the augmentation of its parameter count. This finding suggest that as models increase in size, their performance exhibits a corresponding enhancement.

Table 2: Performance comparison on an arithmetic dataset containing 9,592 test cases between MathGLM and the leading LLMs.

| Model | GPT-4 | ChatGPT | MathGLM-10M | MathGLM-100M | MathGLM-500M | MathGLM-2B |
|---|---|---|---|---|---|---|
| ACC | 18.84% | 10.00% | 61.21% | 70.28% | 89.57% | 93.03% |
| RE | - | - | 97.83% | 99.28% | 99.41% | 99.71% |

**Generalization Analysis.** To assess the generalization ability of MathGLM beyond the 5-digit range, a set of 50,000 training records involving numbers within the 12-digit range are introduced into the training dataset. After incorporating this additional data, MathGLM is further pre-trained for 20,000 steps to enhance its ability to handle arithmetic tasks involving numbers outside the 5-digit range. Table 3 shows the arithmetic performance comparison across various digit ranges, spanning from 5 digit to 12 digit, and involving a mix of arithmetic operations. In comparison to GPT-4 and ChatGPT, our proposed MathGLM consistently achieves the highest accuracy across all digit ranges, indicating the superiority of MathGLM for multi-digit arithmetic operations. A noticeable observation is that a decline in accuracy as the number of digits in the arithmetic operations increases.

Table 3: Performance comparison between most powerful LLMs and MathGLM on various multi-digit arithmetic operations.

| Model | 5-D | 6-D | 7-D | 8-D | 9-D | 10-D | 11-D | 12-D |
|---|---|---|---|---|---|---|---|---|
| GPT-4 | 6.67% | 10.00% | 3.33% | 3.13% | 6.90% | 3.33% | 0% | 6.90% |
| ChatGPT | 5.43% | 2.94% | 1.92% | 1.43% | 1.57% | 1.45% | 0% | 1.33% |
| MathGLM-500M | 83.44% | 79.58% | 71.19% | 64.62% | 66.66% | 49.55% | 42.98% | 27.38% |
| MathGLM-2B | 86.16% | 78.17% | 73.73% | 67.69% | 69.60% | 65.77% | 57.89% | 41.05% |

**Scaling Analysis.** To comprehensively assess the effect of model parameters and training data sizes on performance, we conduct a series of scaling analysis experiments. The model parameters of MathGLM are designed as a range of $\{10M, 100M, 500M, 2B\}$ and the training data sizes is set to a range of $\{1M, 5M, 10M, 25M, 50M\}$. Figure 4 shows the evaluation performance of MathGLM under various scaling configurations. As expected, the performance trend highlights that the 2B model consistently outperforms its smaller counterparts when evaluated using equivalent data sizes, illustrating the positive impact of larger model parameters on arithmetic performance. Besides, it is evident that larger data sizes have a substantial influence on improving the arithmetic performance as well. However, it is important to note that the effect of data size on the smaller model sizes may not be as pronounced as compared to the larger models. This discernible pattern implies that the benefits derived from increasing the data size tend to be more substantial when paired with larger model parameters.

Furthermore, by analyzing the trend illustrated in Figure 4, we attempt to extend our findings and make predictions for scaling configurations that were not directly studied. Employing a log-linear trend assumption, we can extrapolate the results to estimate the requisite model size for achieving a targeted performance when utilizing a more extensive training set. Figure 5 illustrates the extrapolated outcomes derived from the log-linear trend. To validate the validity of this trend, we pre-train a MathGLM equipped with 6B model parameters. From Figure 5, we can observe that the extrapolated trend aligns with the performance achieved by the MathGLM-6B.

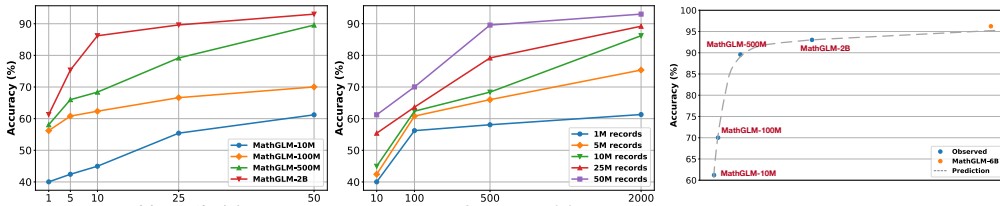

Figure 4: Performance visualization on MathGLM under different scaling configurations, including model param -eters and training data sizes.

Figure 5: The log-linear trend exhibited by the MathGLM. This trend accurately predicts MathGLM-6B's performance.

**Discussions.** Due to the page limit, some additional experiments are reported in Appendix D. Compared with different prominent LLMs including PT-4, ChatGPT, text-davinci-003, code-davinci-002, Galactica, LLaMA, OPT, BLOOM, and GLM, MathGLM consistently achieves superior performance (Cf. Appendix D.1 and Table 10). MathGLM outperforms well-known chat-type LLMs in various arithmetic operations (Cf. Appendix D.2). For BIG-bench arithmetic dataset, MathGLM consistently maintains high accuracy levels even in high-digit arithmetic tasks (Cf. Appendix D.3). Results

on MATH401 can be found in Appendix D.4. A detailed analysis of error categories and their potential causes is reported in Appendix D.5. By leveraging the step-by-step strategy, MathGLM-2B achieves the accuracy raises from 40.76% to 93.03% (Cf. Appendix D.6). The examples generated by MathGLM-2B on various arithmetic tasks are shown in Appendix D.7.

## 3.2 LEARNING ON MATH WORD PROBLEMS

**Results on the Ape210K test dataset.** We report the performance results of various LLMs including GPT-4, ChatGPT, and a series of our MathGLM variations in Table 4. The results show that when paired with GLM-10B, MathGLM achieves performance levels comparable to the state-of-the-art GPT-4 model in terms of answer accuracy. Furthermore, we report the arithmetic accuracy, which measures the correctness of the generated arithmetic expressions. Notably, MathGLM consistently achieves higher arithmetic accuracy compared to answer accuracy across different model sizes. It is obviously observed that augmenting model size tends to bolster its overall performance by comparing Math-GLM's performance with GLM-Large, GLM-6B, and GLM-10B. However, it is worth noting that the performance of MathGLM drops significantly compared to the GLM models when it is coupled with ChatGLM models. A possible explanation is that ChatGLM models are fine-tuned using the instruction data, potentially compromising the inherent capabilities of language models. This tuning process might introduce biases or constraints that hinder the overall ability of the language models in handling math word problems.

Table 4: Performance comparison among different language models on the Ape210K dataset.

| Model | Arithmetic$_{Acc}$ | Answer$_{Acc}$ |
|---|---|---|
| GPT-4 | - | 59.57% |
| ChatGPT | - | 39.78% |
| GLM-Large | - | 0% |
| w/ MathGLM | 62.00% | 50.80% |
| GLM-6B | - | 3.94% |
| w/ MathGLM | 64.60% | 48.06% |
| GLM-10B | - | 0% |
| w/ MathGLM | 69.08% | 58.68% |
| GLM2-6B | - | 31.42% |
| w/ MathGLM | 52.24% | 45.48% |
| ChatGLM-6B | - | 6% |
| w/ MathGLM | 58.52% | 42.28% |
| ChatGLM2-6B | - | 31.70% |
| w/ MathGLM | 50.38% | 43.14% |

**Results on the K6 dataset.** To assess the mathematical problem-solving abilities across different grade levels, we present the performance results on the K6 dataset for various LLMs in Figure 6. A general trend of performance decreases as the grade level increases. Such observation indicates that solving math word problems becomes progressively more challenging for LLMs as the grade level increases, requiring more advanced problem solving skills and a deeper understanding of mathematical concepts. GPT-4 exhibits consistently high accuracy levels across most grade levels, while ChatGPT outperforms the majority of Chinese LLMs across different grade levels. Among the evaluated Chinese LLMs, ChatGLM2-6B demonstrates a commendable level of performance, achieving satisfactory accuracy (reaching 60% accuracy) in solving math word problems from grade 1 to 4. However, its effectiveness diminishes when attempting to solve problems in grade 5 and 6. MathGLM consistently outperforms ChatGPT and many of the most powerful Chinese LLMs from grade 1 to grade 6. Particularly, MathGLM achieves higher accuracy than GPT-4 in more advanced grades, such as grade 5 and 6. This observations show the effectiveness of MathGLM in enhancing the accuracy of solving math word problems, especially in challenging educational contexts that demand deeper mathematical understanding and advanced problem-solving skills. The detailed introduction of these baseline models is provided in Appendix E.

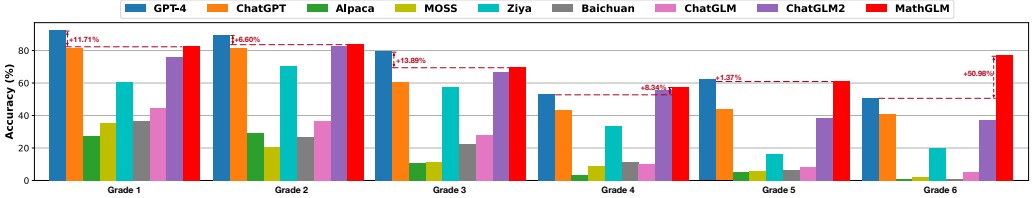

Figure 6: Performance comparison between MathGLM and other popular language models on the K6 dataset.

**Step-by-Step Analysis for MWP.** Figure 7 and Figure 8 demonstrate the performance comparison of MathGLM across different GLM and ChatGLM models respectively. In terms of arithmetic accuracy, as shown in Figure 8, the MathGLM equipped with the step-by-step strategy records marginally lower scores than its counterpart without the strategy. This can be attributed to the fact that the step-by-step approach necessitates a sequential calculation for each mathematical problem. This encourages MathGLM to concentrate on grasping the foundational mathematical rules. Consequently, a portion of the MathGLM's processing power is dedicated to understanding and generating step-by-step solutions, which might slightly weaken its prowess in precisely crafting arithmetic expressions. Nevertheless, while there's a minor dip in arithmetic accuracy, the step-by-step strategy significantly bolsters MathGLM's answer accuracy (Cf. Figure 7). By guiding MathGLM to derive answers progressively, MathGLM generates higher accuracy in solving math word problems. Notably, we observe pronounced improvements in answer accuracy across all GLM variants: 37.86% for GLM-Large, 42.29% for GLM-10B, 47.97% for GLM-6B, and 53.96% for GLM2-6B. Similar trends are also evident in the ChatGLM models, recording gains of 40.65% in ChatGLM-6B and 49.38% in ChatGLM2-6B. These results highlight the inherent trade-off between arithmetic accuracy and answer accuracy by employing the step-by-step strategy. Although this strategy may introduce some potentially impact on arithmetic accuracy, it effectively enhance MathGLM's ability to generate accurate answers for math word problems.

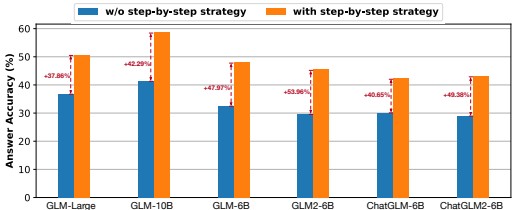

Figure 7: The answer accuracy of MathGLM is compared across various backbone models. A marked improvement in answer accuracy by employing the step-by-step approach.

Figure 8: The arithmetic accuracy of MathGLM is evaluated across various backbone models. A slight decrease in arithmetic accuracy by leveraging the step-by-step strategy.

**Error Distribution.** Here, we construct a percentile graph to analyze the distribution of error types exhibited by the MathGLM on the Ape210K test dataset. As depicted in Figure 9, a prominent error is the "question misunderstood" category. These errors occur when the MathGLM fails to grasp the linguistic nuances and context of specific math word problems, subsequently producing incorrect solutions. Besides, a significant portion of errors is attributed to "calculation error", signaling a need to enhance the computing ability of our language models. In the future, we can further improve the performance of MathGLM based on the type of errors.

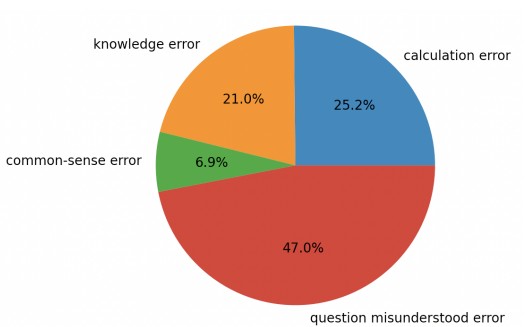

Figure 9: The distribution of error types generated by MathGLM on math word problems.

**Discussions.** Appendix F reports some additional experiments on MWP. The impact of training strategies is reported in Appendix F.1. Performance comparison of MathGLM on MWP among different training dataset sizes and model parameters (Cf. Appendix F.2). A detailed analysis of error categories is reported in Appendix F.3. The impact of training steps is shown in Appendix F.4.

## 4   RELATED WORK

**Arithmetic Calculation.** The emergence of pre-trained Large Language Models (LLMs) (Brown et al., 2020; Chowdhery et al., 2022; OpenAI, 2023) has sparked considerable interest in investigating

their potential for handling arithmetic tasks. Nogueira et al. (2021) and Wang et al. (2021) evaluate the arithmetic capabilities of LLMs on elementary arithmetic operations like addition and subtraction. Muffo et al. (2023) undertake an evaluation that specifically centers on assessing the proficiency of language models in the domain of 2-digit multiplication. BIG-bench (Srivastava et al., 2022) introduces a comprehensive collection of arithmetic datasets, which encompass a spectrum of arithmetic tasks that span numbers within a range of up to 5 digits. Yuan et al. (2023) design an complex arithmetic dataset MATH 401 with various arithmetic operations to evaluate the capabilities of models like GPT-4, ChatGPT, InstructGPT (Ouyang et al., 2022), Galactica (Taylor et al., 2022), and LLaMA (Touvron et al., 2023). To support arithmetic operations involving large numbers, Nye et al. (2021) employ scratchpad-based fine-tuning that enables LLMs to achieve remarkable outcomes in the context of 8-digit addition. Zhou et al. (2022b) adopt the specialize prompt engineering techniques to successfully extend the scope of addition but encountered limitations with multiplication beyond 7 digits. Goat (Liu and Low, 2023) utilizes supervised instruction fine-tuning to handle elementary arithmetic operations with large integers. (Jelassi et al., 2023) investigate length generalization in basic arithmetic tasks via approaches like relative position embeddings and train set priming. Distinguishing itself from these efforts focused on elementary arithmetic, MathGLM pushes the envelope by not only exceeding the realm of basic arithmetic with two numbers but also tackling intricate mixing arithmetic operations involving multiple numbers and diverse data formats. Furthermore, several works explore the integration of external tools for arithmetic tasks. For instance, Toolformer (Schick et al., 2023) adopts an external calculator to accomplish arithmetic calculations, while PoT (Chen et al., 2022) and PAL (Gao et al., 2023) obtain the final answer with the help of programs. Different from leveraging external tools, we focus on explore how to enhance the inherent arithmetic ability of LLMs without relying on external tools.

**Mathematical Reasoning.** LLMs have indeed demonstrated considerable promise in addressing math word problems. (Cobbe et al., 2021) utilize training verifiers to rerank the outputs of LLMs, resulting in remarkable performance on the created GSM8K dataset. Lewkowycz et al. (2022) introduce Minerva, a large language model fine-tuned based on PaLM models (Chowdhery et al., 2022), leveraging a substantial dataset containing scientific and mathematical data. Minerva attains state-of-the-art performance on MATH (Hendrycks et al., 2021) and GSM8K. By leveraging *COT (chain of thought)* (Wei et al., 2022; Kojima et al., 2022; Zhou et al., 2022a) to decompose the math problems into multiple steps, LLMs notably improve their performance in tackling math word problems. Wang et al. (2022) propose the *self-consistency* strategy as a replacement for the decoding strategy used in COT, which brings about better performance than the traditional COT prompting. Uesato et al. (2022) employ process and outcome supervision to enhance the performance of LLMs in solving grade school math problems. Lightman et al. (2023) propose to verify each intermediate reasoning step and find process supervision can significantly improve mathematical reasoning performance. While these studies show the substantial advancements made by LLMs in mathematical reasoning, it is clear that LLMs still make mistakes when confronted with arithmetic operations in math word problems. Different from the aforementioned works that primarily concentrate on improving the reasoning process, our goal is to simultaneously advance both mathematical reasoning and arithmetical calculation capabilities of LLMs, addressing both aspects at the same time.

## 5 CONCLUSION

In this paper, our primary focus revolves around evaluating the mathematical reasoning capabilities of LLMs, encompassing both arithmetic operations and math word problems. For arithmetic tasks, we incorporate step-by-step solution and curriculum learning to train a Transformer-based language model from scratch. With comprehensive training on ample data, we establish that a language model boasting 2 billion parameters can achieve outstanding accuracy in multi-digit arithmetic tasks, exceeding GPT-4's results by a considerable margin. This finding compellingly challenges the prevailing cognition that LLMs face constraints in executing accurate arithmetic operations, especially when dealing with multi-digit numbers, decimals, and fractions, without leaning on external computational aids. When pivoting to math word problems, we reconstruct a dataset enriched with multi-step arithmetic operations. After fine-tuning our MathGLM on this revamped dataset derived from GLM-10B, it achieves similar performance to GPT-4 on the 5,000-sample test set of Chinese math problems, demonstrating its formidable prowess.

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

# A    DATASET DETAILS

## A.1    DATASET ON ARITHMETIC TASKS

**Pre-training Dataset.**    The training dataset for pre-training arithmetic model is created with a Python script. The dataset includes a variety of arithmetic expressions, encompassing different types of arithmetic operations such as addition, subtraction, multiplication, division, and exponentiation. Each expression in the dataset is composed of various types of numbers, including integers, decimals, fractions, percents, and negative numbers. The training dataset consists of approximately 50 million arithmetic sequences. To investigate the impact of dataset scale on the arithmetic performance, we also create multiple datasets of varying sizes, including 1 million, 5 million, 10 million, and 25 million. This diverse representation of numbers ensures that the model can handle a wide range of numerical formats encountered in real-world arithmetic problems.

To facilitate the learning of underlying calculation rules, the arithmetic expressions are designed to be more complex than simple two-number calculations. Instead, each expression in the dataset involves multiple steps of calculations, ranging from 2 to 10 steps. By creating multi-step expressions, the model is exposed to more intricate mathematical reasoning and is better equipped to handle complex arithmetic problem-solving. The details of expressions is presented as follows. Table 5 demonstrates examples from the arithmetic dataset.

- Operations involving integers up to 10,000 that combine addition, subtraction, multiplication, and division.

- Exponentiation tasks using an integer base up to 10,000 and an integer exponent capped at 100.

- Bracketed expressions that include integers up to 10,000, combined with operations such as addition, subtraction, multiplication, and division.

- Lengthy arithmetic expressions that incorporate brackets and blend various numerical types, including integers, decimals, percentages, and negative numbers. These sequences utilize operations such as addition, subtraction, multiplication, and division.

- Arithmetic expressions involving fractions combined with various operations, including addition, subtraction, multiplication, and division.

**Validation Dataset.**    Our evaluation dataset, which comprises 9,592 test cases, is generated from the same distribution as the training dataset, yet remains distinct and is excluded from the training process. This carefully generated suite of datasets serves as a comprehensive benchmark to evaluate and quantify MathGLM's computational prowess across a wide variety of arithmetic tasks.

## A.2    VALIDATION DATASET ON MWP

In the field of math word problems (MWP), the performance of MathGLM is measured using the Ape210K test dataset (Zhao et al., 2020), which contains a collection of 5,000 test math problems. Additionally, we introduce the K6 dataset, which is designed to cover math word problems suitable for elementary school students across 6 different grade levels. The primary purpose of the K6 dataset is to assess the mathematical abilities of LLMs in comprehending and solving general-purpose math reasoning problems. By evaluating MathGLM on the K6 dataset, we are able to gauge its effectiveness in handling mathematical word problems of varying complexity and across a range of grade levels. We collect math word problems from Chinese elementary schools in collaboration with the renowned educational institution, TAL AI Lab. The dataset consists of math problems for each grade level, with each grade containing approximately 100 problems. The wide-ranging nature of these math word problems empowers us to gauge the model's efficacy across an array of difficulty gradients and academic grades. To illustrate the diversity and complexity of the K6 dataset, we present some exemplary math word problems in Table 6. These examples show the range of mathematical concepts covered and the varying levels of difficulty present in the dataset.

Table 5: Examples from the arithmetic dataset where "+", "-", "*", "/", "^" denotes addition, subtraction, multiplication, division, and exponentiation respectively.

| Types | Arithmetic Expression |
|---|---|
| Integre mixing operation | 1+8/1*10+2=1+8*10+2=1+80+2=81+2=83 |
| | 53-2+23+51*56=53-2+23+2856=51+23+2856=74+2856=2930 |
| | 214-792*509*260*556=214-403128*260*556=214-104813280*556=214-58276183680=-58276183466 |
| | 1912*6800*6022-7250-1624=13001600*6022-7250-1624=78295635200-7250-1624=78295627950-1624=78295626326 |
| Exponentiation | 5170^0=1, 1^8756=1 |
| | 3^9=19683, 93^18=270827695297250208363869180422467849 |
| | 100^13=10000000000000000000000000 |
| Expression of fractions | ((49/24)*-(8/70))/-(34/80)=(+(49/24)*(8/70))/(34/80)=(392/1680)/(34/80)=(7/30)/(34/80)=(7/30)*(80/34)=(560/1020)=28/51 |
| | (9947/9276)+(4411/9276)=14358/9276=2393/1546 |
| Expression with brackets | -7805+(4383/7377)=-7805+0.5941439609597398=-7804.405856039048 |
| | 8371*(-1945+8878)=8371*(-1945+8878)=8371*6933=58036143 |
| Lengthy arithmetic expressions | (-2090-5457.35697)*73.0=-7547.35697*73.0=-550957.05881 |
| | -4457+(-7823/5483%)*-3338=-4457+(-7823/54.83)*-3338=-4457+(-142.6773664052526)*-3338=-4457+-142.6773664052526*-3338=-4457+142.6773664052526*3338=-4457+476257.0490607332=471800.0490607332 |

| Grade | Example |
|---|---|
| K1 | 李老师买了20颗糖果,送给小丽5颗,送给小刚8颗,还剩多少颗糖果? |
| K2 | 一个乘数是4,另一个乘数是7,积是多少? |
| K3 | 乐乐家养了36只小鸡,其中1/4是公鸡,母鸡是公鸡的3倍,公鸡和母鸡各有多少只? |
| K4 | 公益小组的同学为敬老院的老人们制作香囊(náng ),12个组共制作了864个,每组都有9人,平均每人制作了几个? |
| K5 | 东、西两城相距180千米,甲、乙两车分别从东、西两城同时出发,相向而行,1.2小时后两车可相遇.实际甲车出发0.4小时后因故障停车,乙车又走了2小时才和甲车相遇,求乙车每小时行多少千米? |
| K6 | 甜甜读一本小说,第一天读了这本书的3/8,正好是180页,第二天又读了这本书的1/6,第2天读了多少页? |

Table 6: Examples from the K6 dataset to demonstrate the diversity and complexity of this dataset.

# B TRAINING DETAILS

## B.1 OVERVIEW OF MATHGLM ON ARITHMETIC TASKS

**Training Parameters for MathGLM.** Table 7 reports an overview of all the models with different model parameters, including hidden dimensions, the number of attention heads, and the total number of layers employed in the model. Besides, we offer detailed training steps to facilitate the reproduction of our MathGLM.

Table 7: Model sizes and architectures of MathGLM.

| Model | Dimension | Heads | Layers | Parameters | Training Steps |
|---|---|---|---|---|---|
| MathGLM-10M | 256 | 32 | 15 | 10M | 120,000 |
| MathGLM-100M | 512 | 32 | 35 | 100M | 155,000 |
| MathGLM-500M | 1024 | 32 | 40 | 500M | 135,000 |
| MathGLM-2B | 2048 | 32 | 40 | 2B | 155,000 |

**Tokenization for Arithmetic Tasks.** The arithmetic operations in our MathGLM involve numbers from 0 to 9, and the calculating signs comprise addition (+), subtraction (-), multiplication (*), division (/), and exponentiation (^). Symbols that represent forms in the data include the decimal point (.), percent sign (%), negative sign (-), fraction delimiter (/), brackets such as '(' and '[', and the equal sign (=). To achieve a consistent tokenization process, we adopt the unified tokenization tool *icetk* proposed in CogView2 (Ding et al., 2022). By leveraging this methodology, we tokenize each digit as a distinct token. For instance, the numeral "12345" is tokenized into the set $\{1, 2, 3, 4, 5\}$. To allocate singular tokens to the other mentioned symbols, we disengage the continuous representation symbols within icetk throughout the tokenization procedure.

Table 8 shows some tokenization examples employed in MathGLM. This tokenization approach ensuers that every element in the arithmetic expression is adequately represented and can be efficiently processed by the MathGLM, facilitating MathGLM to excute comprehensive arithmetic tasks. Owing to the variable lengths of arithmetic expressions, it becomes imperative to standardize their lengths for efficient training of the MathGLM. A straightforward method, like padding each input to a fixed length, might damage training efficacy. To circumvent this, we adopt a more efficient strategy, where multiple arithmetic expressions are concatenated until they achieve a predefined fixed length.

Table 8: Some examples of tokenization in MathGLM.

| Input | Tokenization |
|---|---|
| 12345+345= | ['_', '1', '2', '3', '4', '5', '+', '3', '4', '5', '=']
[20005, 20009, 20010, 20013, 20016, 20015, 20065, 20013, 20016, 20015, 20054] |
| 1234-45678= | ['_', '1', '2', '3', '4', '-', '4', '5', '6', '7', '8', '=']
[20005, 20009, 20010, 20013, 20016, 20011, 20016, 20015, 20021, 20025, 20023, 20054] |
| 34*678= | ['_', '3', '4', '*', '6', '7', '8', '=']
[20005, 20013, 20016, 20032, 20021, 20025, 20023, 20054] |
| 1.2/2= | ['_', '1', '.', '2', '/', '2', '=']
[20005, 20009, 20007, 20010, 20026, 20010, 20054] |
| (1.2*3%)/2+[(12+3)*5]= | ['_', '(', '1', '.', '2', '*', '3', '%', ')', '/', '2', '+', '[', '(', '1', '2', '+', '3', ')', '*', '5', ']', '=']
[20005, 20020, 20009, 20007, 20010, 20032, 20013, 20040, 20014, 20026, 20010, 20065, 20052, 20020, 20009, 20010, 20065, 20013, 20014, 20032, 20015, 20042, 20054] |

## B.2 BACKBONE MODELS

General Language Model (GLM) is a Transformer-based language model that combines autogressive blank infilling with bidirectional attention mechanisms. Different from decoder-only language models that primarily rely on unidirectional attention, GLM integrates bidirectional attention on unmasked contexts. This innovative approach empowers it with heightened proficiency in both comprehension and generative tasks.

**Pre-Training Objectives.** To amplify its linguistic understanding and generative abilities, GLM incorporates a dual pre-training strategy: 1) *Autoregressive Blank Infilling* involves predicting missing tokens within spans of corrupted text, wherein segments are arbitrarily supplanted with a [MASK] token. 2) *Multi-Task Pretraining* is utilized to endow GLM text generation ability, which aims to generate longer text by sampling random-length span from document-level or sentence-level text.

**Model Sizes.** GLM offers a diverse of models with various model parameters, including GLM-Large, GLM-6B, GLM-10B, GLM2-6B, ChatGLM-6B, and ChatGLM2-6B. Comprehensive specifics concerning the hyperparameters for each model variant can be found in Table 9. GLM-Large model is specifically tailored for Chinese language processing tasks equipped with 335M model parameters, while GLM-10B, GLM-6B, and GLM2-6B are equipped with 10 billion, 6 billion, and 6 billion parameters, respectively, enabling them to handle a wide range of NLP tasks with varying complexities. Augmenting the series are bilingual conversational models: ChatGLM-6B and ChatGLM2-6B, both tailored for Chinese-English bilingual dialogue tasks. The ChatGLM-6B model, having 6.2 billion parameters, undergoes fine-tuning using Chinese Q&A and dialogue datasets. In contrast, ChatGLM2-6B emerges as an evolved iteration of ChatGLM-6B, marking enhancements in performance, extended context handling, optimized inference, and broader applicability.

Table 9: Hyperparameters of the backbone models.

| Model | Dimension | Heads | Layers | Parameters |
|---|---|---|---|---|
| GLM-Large | 1024 | 24 | 16 | 335M |
| GLM-10B | 4096 | 64 | 48 | 10B |
| GLM-6B | 4096 | 32 | 28 | 6.2B |
| GLM2-6B | 4096 | 32 | 28 | 6.2B |
| ChatGLM-6B | 4096 | 32 | 28 | 6.2B |
| ChatGLM2-6B | 4096 | 32 | 28 | 6.2B |

## C EVALUATION METRIC

To measure the ability of MathGLM on arithmetic tasks, we adopt the following metrics to evaluate the outputs.

**Accuracy** is typically measured by comparing the output of the MathGLM and the ground truth answer. In our experiments, we adhere to standard rounding rules, constraining the generated answers to precisely two decimal places. When the correctly rounded answer aligns with the answer generated by the MathGLM, we classify this outcome as a correct answer.

**Relative Error** is another important metric used to evaluate the effectiveness of MathGLM, which quantifies the difference between the output generated by MathGLM and the correct answer. The relative error (RE) is quantified using the following formula:

$$RE = |\frac{\hat{y} - y}{y}| \tag{1}$$

where $\hat{y}$ and $y$ denote the generated answer and the correct answer respectively. For our evaluation purposes, we utilize a relative error threshold of 1%. This threshold serves as a criterion for determining the acceptability of the answers generated by the MathGLM, where any relative error falling within this threshold range is considered an accurate outcome.

# D ADDITIONAL EXERIMENTS ON ARITHMETIC TASKS

## D.1 RESULTS ON TEST-100

Additionally, we conduct a performance comparison of arithmetic tasks among different prominent large language models (LLMs) including GPT-4, ChatGPT, text-davinci-003, code-davinci-002, Galactica, LLaMA, OPT, BLOOM, and GLM. For this comparison, we randomly extract a compact arithmetic dataset *Test-100* containing 100 test cases from the larger dataset discussed earlier. The results of this comparison arithmetic performance are presented in Table 10. Upon analyzing the results, it is evident that MathGLM achieves a high accuracy of 93.03% with 2 billion model parameters, surpassing all other LLMs. In addition to leading models like GPT-4 and ChatGPT, the large science model Galactica exhibits better performance in arithmetic tasks. This can be attributed to Galactica's training on a large scientific corpus, enabling it to learn the languages of science and comprehend the intricacies of arithmetic tasks. By leveraging the unique characteristics of this dataset, Galactica is able to enhance its understanding and handling of arithmetic tasks, resulting in improved performance. These findings emphasize the significance of domain-specific training and leveraging specialized datasets to enhance model performance. Besides, a step-by-step solution strategy, which involves decomposing complex arithmetic expressions into individual steps,

Table 10: Overall performance comparison on various LLMs in term of Accuracy.

| Model | ACC | RE |
|---|---|---|
| GPT-4 | 22.22% | - |
| ChatGPT | 13.25% | - |
| text-davinci-003 | 9.79% | - |
| text-davinci-002 | 4.08% | - |
| Galactica-120b | 7.97% | - |
| Galactica-30b | 7.02% | - |
| LLaMA-65b | 5.02% | - |
| OPT-175B | 3.83% | - |
| BLOOM-176B | 3.96% | - |
| GLM-130B | 3.06% | - |
| MathGLM-10M | 64.29% | 97.96% |
| MathGLM-100M | 73.47% | 98.23% |
| MathGLM-500M | 89.80% | 98.82% |
| MathGLM-2B | 94.90% | 98.98% |

has proven to be effective in improving arithmetic performance. The outstanding performance of MathGLM shows that the language model coupled with a specialized dataset and the step-by-step solution strategy can achieve remarkable performance in arithmetic tasks.

## D.2 GROUPED RESULTS

To clearly evaluate the arithmetic ability of MathGLM among different operations, we design a series of extended experiments. Specifically, we design small test datasets comprising 100 test cases to respectively evaluate the arithmetica performance of MathGLM in various arithmetic operations, including addition, subtraction, multiplication, and division. These datasets encompass different data formats, such as integers, decimals, percents, fractions and negative numbers. Here, we compare MathGLM with several well-known chat-type LLMs, such as GPT-4, ChatGPT, ChatGLM, and Bard. The arithmetic performance comparison among these different language models is demonstrated in Table 11. Analyzing the results, we can observe that the majority of LLMs exhibit commendable accuracy levels exceeding 90% across diverse data formats for elementary arithmetic operations like addition and subtraction. However, as the complexity escalates to operations like multiplication and division, a divergence in performance manifests across different models. For instance, the accuracy levels of the most powerful model GPT-4 also show a trend towards zero, especially when dealing with decimal and percentile data formats. In contrast, MathGLM consistently shows superior performance in multiplication operations across various data formats, surpassing the capability of GPT-4. This demonstrates the effectiveness and capabilities of MathGLM in handling complex arithmetic tasks, even outperforming a prominent model like GPT-4 in specific operations. Notably, even the smaller variant of MathGLM, MathGLM-10M, with only 10 million training parameters, also achieves remarkable arithmetic performances, further emphasizing the arithmetic capabilities of our MathGLM.

## D.3 RESULTS ON BIG-BENCH

We also evaluate MathGLM using BIG-bench arithmetic dataset (Srivastava et al., 2022), which is commonly used to evaluate basic arithmetic capabilities of language models by performing n-digit addition (ADD), subtraction (SUB), multiplication (MUL), and division (DIV). Table 12 reports

| Task | Format | GPT-4 | ChatGPT | ChatGLM | Bard | MathGLM-10M | MathGLM-2B |
|------|--------|-------|---------|---------|------|-------------|------------|
| ADD | Int | 100% | 100% | 94% | 96.0% | 100% | 100% |
|     | Dec | 100% | 98% | 76% | 87% | 96% | 100% |
|     | Frac | 43.33% | 17.02% | 32.98% | 14.2% | 60.64% | 100% |
|     | Perc | 100% | 90.0% | 1% | 9.6% | 100% | 100% |
|     | Neg | 100% | 98% | 91% | 95% | 100% | 100% |
| SUB | Int | 100% | 97% | 89% | 91% | 98% | 100 % |
|     | Dec | 100% | 94% | 82% | 85% | 98% | 100% |
|     | Frac | 52.48% | 18.81% | 3% | 24.24% | 68.32% | 96.04% |
|     | Perc | 100% | 100% | 18% | 0% | 99% | 100% |
|     | Neg | 100% | 97% | 44% | 78% | 100% | 100% |
| MUL | Int | 9% | 4% | 1% | 2% | 77% | 84% |
|     | Dec | 0% | 0% | 0% | 0% | 3% | 33% |
|     | Frac | 5.63% | 2.82% | 1.41% | 1.41% | 67.61% | 85.92% |
|     | Perc | 0% | 0% | 1% | 0% | 81% | 97% |
|     | Neg | 7% | 2% | 0% | 0% | 76% | 98% |
| DIV | Int | 92% | 91% | 24% | 68% | 99% | 100% |
|     | Dec | 93% | 88% | 60% | 60% | 97% | 98% |
|     | Frac | 33.44% | 29.69% | 7.81% | 1.56% | 73.44% | 96.88% |
|     | Perc | 97% | 80% | 19% | 15% | 88% | 100% |
|     | Neg | 97% | 90% | 50% | 52% | 96% | 100% |

Table 11: Arithmetic comparison between MathGLM and other LLMs among different operations. Int denotes integers, Dec denotes decimals, Frac denotes fractions, Perc denotes percents, and Neg denotes negative numbers.

the experimental results of GPT-4 and MathGLM on various arithmetic operations with different numbers of digits. GPT-4 exhibits near-perfect (100%) accuracy in low-digit arithmetic tasks. However, as the digits escalate, the performance gradually diminishes, particularly pronounced in the multiplication task. In contrast, MathGLM consistently maintains high accuracy levels even in high-digit arithmetic tasks, illustrating its outstanding ability to handle complex arithmetic tasks effectively. The performance trends of different MathGLM variants reveal a consistent pattern of improvement as model size increases. For ADD and SUB tasks, the accuracy remains consistently high across all model sizes with slight variations. There is a tendency for larger models to achieve higher accuracy compared to smaller models but the differences in performance between different model sizes are relatively small. In the MUL task, accuracy rises distinctly with larger model sizes. Smaller models exhibit relatively lower accuracy, while larger counterparts demonstrate enhanced accuracy, particularly in tasks involving higher digit numbers. A similar tendency can be observed in the DIV task. Overall, the evaluation results demonstrate that MathGLM outperforms GPT-4 in high-digit arithmetic tasks, and the performance generally inproves with larger model sizes.

### D.4 RESULTS ON MATH 401

Table 13 shows a comprehensive evaluation of the arithmetic performance of MathGLM on the MATH 401 dataset (Yuan et al., 2023). This dataset offers a new set of arithmetic problems, allowing for a deeper exploration into MathGLM's proficiency in addressing a wide variety of arithmetic tasks. By evaluating MathGLM's performance on this dataset, we observe that MathGLM consistently outperforms all other large language models with a substantial number of model parameters.

### D.5 ANALYSIS ON ARITHMETIC ERRORS

Despite achieving an impressive overall accuracy of 93.03% with its 2 billion model parameters, a thorough analysis is conducted to comprehend instances where MathGLM fails to generate accurate answers. Consider the example $3468 * 4046/7424$, MathGLM generate an answer of $468 * 4046/7424 = 14031528/7424 = 1889.901400862069$, while the true answer is

Table 12: Overall performance comparison on GPT-4 and MathGLM on BIG-bench Arithmetic sub-task.

| Task | | | GPT-4 | MathGLM-10M | MathGLM-100M | MathGLM-500M | MathGLM-2B |
|---|---|---|---|---|---|---|---|
| ADD | 1D | | 100% | 84% | 100% | 100% | 100% |
| | 2D | | 100% | 97.2% | 100% | 100% | 100% |
| | 3D | | 99.6% | 99.3% | 100% | 100% | 100% |
| | 4D | | 98.8% | 99.9% | 99.9% | 100% | 100% |
| | 5D | | 94.1% | 99.2% | 100% | 99.6% | 99.4% |
| SUB | 1D | | 100% | 92% | 100% | 100% | 100% |
| | 2D | | 100% | 98.5% | 99.8% | 100% | 100% |
| | 3D | | 99.2% | 98.8% | 99.9% | 100% | 99.9% |
| | 4D | | 98.9% | 98.4% | 99.6% | 99.7% | 99.8% |
| | 5D | | 92.4% | 98.0% | 99.3% | 99.5% | 98.9% |
| MUL | 1D | | 100% | 91% | 100% | 99% | 100% |
| | 2D | | 99.4% | 85.8% | 99.7% | 99.9% | 99.9% |
| | 3D | | 30.3% | 77.8% | 91.4% | 93.7% | 98.3% |
| | 4D | | 5.3% | 79.7% | 80.4% | 90.0% | 94.9% |
| | 5D | | 0.0% | 41.6% | 55.6% | 59.6% | 89.9% |
| DIV | 1D | | 100% | 87.0% | 100% | 100% | 100% |
| | 2D | | 100% | 89.5% | 100% | 100% | 100% |
| | 3D | | 94.5% | 90.2% | 100% | 99.6% | 99.4% |
| | 4D | | 90.9% | 90.5% | 99.5% | 99.6% | 100% |
| | 5D | | 53.4% | 82.2% | 92.9% | 93.6% | 94.9% |

$468 * 4046/7424 = 14031528/7424 = 1890.0226293103$. Upon comparing the generated results with the true answers, it is obviously observed that the multiplication operation for $468 * 4046$ is correct but the division operation for $14031528/7424$ is incorrect. One possible reason for this discrepancy is that MathGLM's pre-training primarily encompasses numbers in the 5-digit range, thereby causing inaccuracies when tackling division tasks involving 12-digit and 4-digit numbers. Upon thorough analysis of the errors made by MathGLM, it's important to highlight that the inaccuracies in the generated answers are remarkably close to the correct evaluations.

Table 14 provides some examples to analyze the failures of MathGLM on performing arithmetic tasks. Through careful examination of these examples, we can observe several patterns and trends in the MathGLM's errors. Firstly, MathGLM appears to grapple with intricate arithmetic expressions, particularly those combining several operations and large numbers. For instance, the expression 14031528/742: the division of an 8-digit number by a 4-digit one proves problematic for MathGLM, leading to miscalculations in the outcome. Secondly, MathGLM tends to encounter difficulties when dealing with long sequences of numbers and operations. As the expression length increases, the model's ability to accurately perform arithmetic calculations diminishes, leading to inaccurate results. For example, expression involving multiplication among two large numbers like 3626 * 8919 and calculation with a decimal and large integer number like 1.610311 * 7691. These errors generated by MathGLM usually have only one calculation result error, indicating that the MathGLM's mistakes mainly occur at specific calculation steps rather than affecting the entire expression.

## D.6   STEP-BY-STEP ANALYSIS

To delve deeper into the impact of the step-by-step strategy on MathGLM, we conduct extended experiments that directly calculate the answer of each arithmetic expression without employing the step-by-step approach. Figure 10 shows performance comparison between employing the step-by-step strategy and bypassing it for different models. We can observe that a significant improvement in the peformance of MathGLM when the step-by-step strategy is applied. For instance, in the case of MathGLM-500M, the accuracy rises from 31.96% to 89.57%, while for MathGLM-2B, it increases

Table 13: Overall performance comparison on various LLMs in term of Accuracy.

| Model | ACC |
|---|---|
| GPT-4 | 83.54% |
| GPT-3.5-turbo | 75.06% |
| text-davinci-003 | 56.61% |
| text-davinci-002 | 42.89% |
| code-davinci-002 | 21.70% |
| Galactica-120b | 45.14% |
| Galactica-30b | 45.14% |
| Galactica-6.7b | 34.41% |
| LLaMA-65b | 28.43% |
| LLaMA-30b | 30.17% |
| LLaMA-13b | 27.68% |
| LLaMA-7b | 21.96% |
| OPT-175B | 21.70% |
| OPT-66B | 20.70% |
| OPT-30B | 15.96% |
| OPT-13B | 15.21% |
| OPT-6.7B | 14.46% |
| BLOOM-176B | 22.44% |
| BLOOM-7.1B | 7.23% |
| BLOOM-3B | 4.24% |
| BLOOM-1.7B | 5.24% |
| GLM-130B | 25.94% |
| GLM-10B | 14.96% |
| MathGLM-0.5B | 85.48% |
| MathGLM-2B | 89.44% |

Table 14: Some failed examples generated by MathGLM.

| Input | Output | Ground Truth |
|---|---|---|
| 3468*4046/7424= | 14031528/7424=1889.901400862069 | 14031528/7424=1890.0226293103449 |
| (3626*8919)/8861= | 32330294/8861=3648.605574991536 | 32340294/8861=3649.7341157882856 |
| 7715/4791*7691-1968*9155= | 1.610311*7691-1968*9155=12384.801801-1968*9155=12384.801801-18017040=-18004655.198199 | 1.610311*7691-1968*9155=12384.9018993-1968*9155=12384.9018993-18017040=-18004655.098100606 |
| (4059+7011.8718)-4038.22*847.15907= | (4059+7011.8718)-4038.22*847.15907=11070.8718-4038.22*847.15907=11070.8718-3420014.6996554=-3408943.8278554 | (4059+7011.8718)-4038.22*847.15907=11070.8718-4038.22*847.15907=11070.8718-3421014.6996554=-3409943.8278554003 |
| 7499-5747.91007/-5438*-439= | 7499-5747.91007/5438*439=7499-1.0570081040823832*439=7499-464.0265576921662=7034.973442307834 | 7499-5747.91007/5438*439=7499-1.056989715*439=7499-464.0184848713=7034.981515128724 |
| 3868*6735*5755+3741-7533= | 26050980*5755+3741-7533=159923389900+3741-7533=159923393641-7533=159923386108 | 26050980*5755+3741-7533=149923389900+3741-7533=149923393641-7533=149923386108 |

from 40.76% to 93.03% for MathGLM-2B, all attributable to the incorporation of the step-by-step strategy. Similarly, the relative error accuracy exhibits a similar positive trend, escalating from 89.29% to an exceptional 99.41% for MathGLM-500M, and from 94.26% to an outstanding 99.71% for MathGLM-2B with the implementation of the step-by-step strategy. These results demonstrate the effectiveness of the step-by-step strategy in enhancing MathGLM's ability to accurately perform arithmetic operations. The step-by-step approach enables MathGLM to better understand and solve intricate arithmetic tasks, leading to significant improvements in accuracy and relative error accuracy metrics.

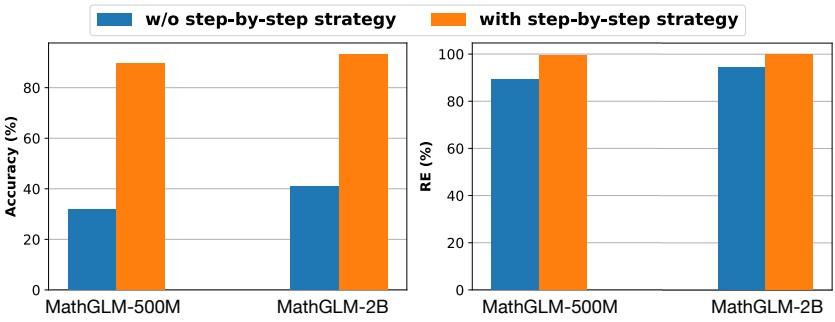

Figure 10: Performance comparison of MathGLM with and without the step-by-step solution.

## D.7 CASE STUDY

Figure 11 demonstrates examples generated by MathGLM with 2B model parameters on addition, subtraction, multiplication, division, exponentiation, and mixing operations tasks.

---

**MathGLM-2B**

**Addition**

5678+9865=
5678+9865=15543

67548+908653=
67548+908653=976201

8659054861+7564369655=
8659054861+7564369655=16223424516

7878.547+98065.344901=
7878.547+98065.344901=105943.891901

176.67+2.3%=
176.67+2.3%=176.67+0.23=176.9

(13/14)+(18/19)=
(13/14)+(18/19)=247/266+252/266=499/266

-32+7854=
-32+7854=7822

**Subtraction**

76869-98654=
76869-98654=-21785

46455762897-9867553=
46455762897-9867553=46445895344

879800.75432-767.427934=
879800.75432-767.427934=879033.326386

-7634-8904=
-7634-8904=-16538

5423%-638.0965=
5423%-638.0965=54.23-638.0965=-583.8665

(3/5)-(12/19)=
(3/5)-(12/19)=57/95-60/95=-3/95

21-(19/21)=
21-(19/21)=21-0.9047619047619048=20.095238095238

**Multiplication**

1234*7809=
1234*7809=9636306

8757755*907644=
8757755*907644=7948923779220

9765.89755*42653.9773065=
9765.89755*42653.9773065=416554372.4753039

-453*75479=
-453*75479=-34191987

7645%*975301=
7645%*975301=76.45*975301=74561761.45

(123/897)*(6753/16)=
(123/897)*(6753/16)=(830619/14352)=276873/4784

875*(646/67)=
875*(646/67)=875*9.64179104477612=8436.567164179

**Division**

7862/87=
7862/87=90.36781609195402

6574389173/86532=
6574389173/86532=75976.3922363981

8753.75255/5427.67397=
8753.75255/5427.67397=1.6128000040872018

456%/243=
456%/243=4.56/243=0.01876543209876543

(12/33)/(56/12)=
(12/33)/(56/12)=(12/33)*(12/56)=(144/1848)=6/77

-129/-334=
-129/-334=129/334=0.3862275449101796

**Exponentiation**

756^0=
756^0=1

457^2=
457^2=208849

786^1=
786^1=786

12^4=
12^4=20736

386^5=
386^5=8569125814176

723^10=
723^10=3902860000167580990810750464 9

**Mixing Operations**

61+9104+6129-8755-9978*1335*2454-5404=
61+9104+6129-8755-9978*1335*2454-5404
=61+9104+6129-8755-13320630*2454-5404
=61+9104+6129-8755-32688826020-5404
=9165+6129-8755-32688826020-5404
=15294-8755-32688826020-5404
=6539-32688826020-5404
=-32688819481-5404
=-32688824885

8723%/-632+-169.63=
8723%/-632+-169.63=87.23/-632+-169.63
=-87.23/632-169.63=-0.13802215189873418-169.63
=-169.76802215189873

(-2714.9607*4215%)--7850=
(-2714.9607*4215%)--7850=(-2714.9607*42.15)--7850
=(-114435.593505)--7850=-114435.593505--7850
=-114435.593505+7850=-106585.593505

Figure 11: Examples of MathGLM's response on a variety of arithmetic tasks.

# E   BASELINE MODELS FOR MWP

Here, we leverage a variety of popular LLMs that can address Chinese problems to compare the mathematical reasoning ability among these LLMs and our MathGLM. The details of each baseline LLM as follows.

- GPT-4 (OpenAI, 2023) is the most advanced generative language model that developed by OpenAI, which successfully achieves so many SOTA performances on a variety of downstream tasks.

- ChatGPT (OpenAI) is the predecessor of GPT4 and is constructed upon the success of InstructGPT (Ouyang et al., 2022), which is fine-tuned using instruction data with reinforcement learning from human feedback (RLHF), making it a powerful tool for natural language understanding and conversation.

- MOSS (Sun and Qiu) is an open-source LLM that consists of 16 billion model parameters. It utilizes 100 billion Chinese tokens and 20 billion English tokens to learn language patterns and semantic representations.

- Ziya-LLaMA-13B (Zhang et al., 2022a) is a language model constructed on LLaMA-13B, which extends LLaMA-13B's character set to contain 7,000 Chinese characters and undergoes continual pre-training on a vast dataset of 110 billion Chinese tokens.

- Chinese-Alpaca-13B (Cui et al., 2023) is a Chinese language model with 13 billion parameters that is built upon LLaMA-13B. During the supervised instruction tuning, the Low Rank Adaptation (LoRA) (Hu et al., 2021) technique is utilized to fine-tune LLaMA-13B for Chinese language tasks.

- Baichuan-7B (inc.) shares similarities with LLaMA but is pre-trained from scratch on a massive dataset containing 1.2 trillion Chinese and English tokens.

- ChatGLM-6B (THUDM, a) and its successor ChatGLM2-6B (THUDM, b) are language models that share a unified transformer architecture named GLM (Du et al., 2021; Zeng et al., 2022). These models are pre-trained on a diverse dataset containing English and Chinese data, combined with the supervised instruction tuning, makes them powerful tools for understanding and generating text in both English and Chinese contexts.

# F   ADDITIONAL EXPERIMENTS ON MWP

## F.1   COMPARISON OF TRAINING STRATEGIES

Here, we evaluate the mathematical reasoning ability of MathGLM with different training strategies: fine-tuning and continue training. To execute continue training, we amalgamate the Ape210K train dataset with instruction data released by Chinese-Vicuna (Chenghao Fan and Tian, 2023). We subsequently continue training MathGLM from the GLM-10B backbone. Table 15 shows the overall performance comparison of MathGLM employing different training strategies. We observe that directly fine-tuning on the specific dataset can achieves better performance.

Table 15: Overall performance comparison on various LLMs in term of Accuracy.

| Training | w/o step-by-step strategy | | with step-by-step strategy | |
|---|---|---|---|---|
| | Arithmetic$_{Acc}$ | Answer$_{Acc}$ | Arithmetic$_{Acc}$ | Answer$_{Acc}$ |
| Fine-tuning | 71.38% | 41.24% | 69.08 % | 58.68% |
| Continue training | 70.16% | 40.34% | 67.02% | 56.60% |

## F.2   SCALING ANALYSIS

To explore the impact of scaling on MathGLM, we conduct a series of experiments encompassing varying dataset sizes and distinct model parameters. Table 16 demonstrates the results obtained from varying the dataset sizes within the range of $\{5K, 10K, 20K, 50K, 100K, 200K\}$. Furthermore,

to understand the impact of different model parameters, we incorporate various backbone models into MathGLM, including GLM-Large (335M), GLM-6B, and GLM-10B. The results consistently indicate that MathGLM's performance improves across all backbone models with the increase in dataset size. Such observation highlights the beneficial effects of enlarging the training data on bolstering MathGLM's proficiency in tackling math word problems. By accessing more extensive datasets, MathGLM is introduced to a wider array of problem types, resulting in better performance. Additionally, discernible differences in performance emerge among the various backbone models. Given sufficient dataset size, larger models like MathGLM-GLM-10B often outperform others, indicating the crucial role of model parameters in addressing intricate math word problems. These insights emphasize the significance of both dataset and model scaling. By augmenting dataset size and utilizing larger models, we can markedly boost MathGLM's capability to generate more accurate solutions, enhancing its overall efficacy in resolving math word problems.

Table 16: Performance comparison of MathGLM on different training dataset sizes and model parameters.

| Model Scale | MathGLM-GLM-Large | MathGLM-GLM-6B | MathGLM-GLM-10B |
|---|---|---|---|
| 5K Problems | 4.32% | 12.84% | 3.68% |
| 10K Problems | 7.14% | 19.78% | 6.36% |
| 20K Problems | 10.36% | 21.89% | 9.62% |
| 50K Problems | 18.32% | 26.40% | 16.78% |
| 100K Problems | 25.98% | 31.44% | 22.20% |
| 200K Problems | 35.68% | 34.00% | 38.10% |

### F.3 FAILURE ANALYSIS ON MATH WORD PROBLEMS

Figure 12 provides some failed examples generated by MathGLM-GLM-10B on solving math word problems. We can identify certain challenging scenarios where MathGLM-GLM-10B encounters difficulties in solving math word problems. One common issue is the misinterpretation of ambiguous language, leading to incorrect problem-solving approaches. For instance, ambiguous phrases such as "more than" or "less than" can be interpreted differently by the model, resulting in inaccurate solutions. Additionally, MathGLM-GLM-10B tends to struggle with problems that involve complex mathematical operations. As a result, it may provide partially correct arithmetic solutions but fail to arrive at the final correct answer.

### F.4 TRAINING STEPS ANALYSIS.

We explore the impact of training steps on the MathGLM's performance by analyzing its performance against varied training steps, as depicted in Figure 13. The results reveal that there is a consistent uptrend in performance as the number of training steps increases. With more training steps, MathGLM becomes increasingly adept at comprehending and resolving math word problems, which translates to a surge in accuracy. However, it is clearly observed that the performance gains of MathGLM start to plateau after a certain point, indicating potential diminishing returns with extended training. These findings highlight the significance of finding an optimal balance between training time and performance gains for MathGLM in solving math word prblems. Additionally, we observe that model undergoing instruction tuning requires a longer training duration to achieve consistent accuracy on math word problems.

### F.5 CASE STUDY

Here, we present specific cases to demonstrate the solving process of the MathGLM on both arithmetic tasks and math word problems. As shown in Figure 14, these examples illustrate how the MathGLM leverages a step-by-step strategy to solve both arithmetic tasks and math word problems accurately.

For arithmetic tasks, the MathGLM breaks down complex calculations into multiple steps rather than a direct answer. The step-by-step strategy ensures that each intermediate result is accurately computed, leading to the final correct answer for each arithmetic expression.

| Type | Question | Generated Answer | Ground Truth |
|---|---|---|---|
| Question Misunderstood Error | 小红看一本小说，第一天看了总页数的25\%少17页，第二天看的比总页数的(1/8)还多16页，还剩下61页没看，这本小说一共有多少页？ | x=(17+16+61)/(1−25%−(1/8))=(17+16+61)/(1−0.25−(1/8))=(33+61)/(1−0.25−(1/8))=94/(1−0.25−(1/8))=94/(1−0.25−0.125)=94/(0.75−0.125)=94/0.625=144. | x=(61−17+16)/(1−(1/8)−25%)=96 |
| | 牛牛用10元钱买了一个玩具，胖胖对他的玩具非常喜欢，花20元钱买了过去. 过了几天，牛牛有些舍不得，又花30元钱从胖胖手里买了回来. 又过了几天，胖胖花40元又把玩具从牛牛手里买走. 在整个过程中，牛牛共赚了多少元钱？ | x=20+30+40=50+40=90 | x=40−30+20−10=20 |
| Calculation Error | 小马虎在甲数一道减法算式时，错把减数65看成了56，结果得到差是215，正确的差是多少？ | x=215+(56−65)=215+9=222 | x=215+56−65=206 |
| | 学校计划装修16间教室，工程队经过测算，每间教室需要铺地砖192块，工程队应该购买这样的地砖多少块？ | x=1920*16=32640 | x=192*16=3072 |
| Knowledge Error | 一个长方形的周长是88cm，长是30cm，它的面积是多少cm**2？ | x=30*88/(30*2)=30*88/60=2640/60=44 | x=30*(88/2−30)=420 |
| | 一个长方体的长和宽都是4米，高是5米，如果底面积扩大5倍，要使体积不变，高应该是多少厘米？ | x=(4*5*5)/5=(20*5)/5=100/5=20 | x=(5/5)*100=100 |
| Common-sense Error | 两地相距13千米，甲乙二人从两地同时出发相向而行，经过(4/3)小时相遇. 甲每小时行5千米，乙每小时行多少千米？ | x=(13−5*4/3)/(4/3)=4.75 | x=(13−5*4/3)/4=(13−20/3)/4=(13−6.666666666666667)/4=6.333333333333334/4=1.66666667 |
| | 一艘客轮往返甲、乙两港，顺水速度是15千米/小时，逆水速度是的12千米/小时. 现在甲港放一个木排顺水漂流到乙港，要用3天才能到达. 那么，甲、乙两港的水路长多少千米. | x=((15−12)/2)*72=108 | x=(15+12)*3=27*3=81 |

Figure 12: Some failed examples generated by MathGLM-GLM-10B on solving math word problems.

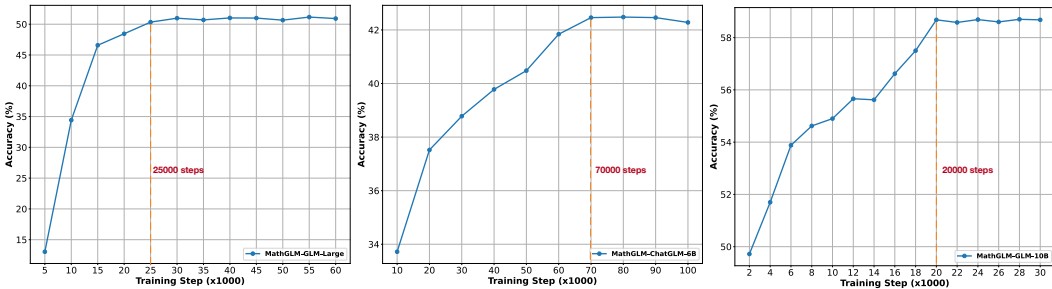

Figure 13: The impact of training steps on MathGLM with different backbone models. Fine-tuning on model undergoing instruction tuning requires a longer training steps.

In math word problems, the MathGLM utilizes LLMs's language understanding capabilities to understand the problem statement accurately. The MathGLM then applies the step-by-step strategy to seperately calculate the generated arithmetic expressions for each math problem, enabling it to obtain the correct answer for math word problems.

| Math Word Problems | | Arithmetic Tasks | |
|---|---|---|---|
| Input | Output | Input | Output |
| 王艳家买了一台洗衣机和一台电冰箱,一共花了6000元,电冰箱的价钱是洗衣机的(3/5),求洗衣机的价钱. | 答:x=6000/(1+(3/5))=6000÷(1+3/5)=6000÷(5/5+3/5)=6000÷8/5=30000/8=3750 | -7439+(3156+1649%)+(-2307+-1112)= | -7439+(3156+16.49)+(-2307+-1112)=-7439+3172.49+(-2307+-1112)=-7439+3172.49+(-2307-1112)=-7439+3172.49+-3419=-7439+3172.49-3419=-4266.51-3419=-7685.51 |
| 甲乙两车从相距180千米的两地同时相向而行,甲车每小时行50千米,乙车每小时行40千米,几小时后两车相遇? | 答:x=180/(50+40)=180/90=2 | | |
| 两个因数相乘,其中一个因数扩大10倍,要使积不变,另一个因数要缩小为原来的多少? | 答:x=(1/10) | 511 4392.7*1127/[(6532--3588)+-1499.90071*-3871.5083]= | 4392.7*1127/[(6532+3588)+-1499.90071*-3871.5083]=4392.7*1127/[10120+-1499.90071*-3871.5083]=4392.7*1127/(10120+1499.90071*3871.5083)=4392.7*1127/(10120+5806878.047940893)=4392.7*1127/5816998.047940893=4950572.899999999/5816998.047940893=0.8510528728389893 |
| 一个长方体的体积是48m**3,长是8m、宽是5m、高是多少m? | 答:x=48/(8*5)=48/40=1.2 | | |
| 李枫看一本书,第一周看了全书的45%,第二周看了全书的(2/5),还有60页没有看.这本书一共有多少页? | 答:x=60/(1-(45%+(2/5))=60/(1-(0.45+(2/5)))=60/(1-(0.45+0.4))=60/(1-0.85)=60/0.15=400 | | |
| 教室长8m、宽6m、高3m,六1班有48名学生,平均每人占有的空间是多少? | 答:x=8*6*3/48=48*3/48=144/48=3 | 2824/2202-[8553-(156-4493)/9780*4740]*8637= | 2824/2202-[8553--4337/9780*4740]*8637=2824/2202-(8553+4337/9780*4740)*8637=2824/2202-(8553+0.4434560327198364*4740)*8637=2824/2202-(8553+2101.9815950920245)*8637=2824/2202-10654.98159509202 5*8637=1.2824699818346957-10654.981595092025*8637=1.2824699818346957-92027076.03680982=-92027074.75433984 |
| 加工一批零件,甲单独做要10天完成,乙单独做要8天完成.现在甲、乙两人合作,多少天能完成这些零件的一半? | 答:x=(1/2)/((1/10)+(1/8))=1/2÷(1/10+1/8)=1/2÷(4/40+5/40)=1/2÷9/40=40/18=20/9 | | |
| 一种商品原价780元,如打八折出售,那么价格降价多少? | 答:x=780*(1-80%)=780*(1-0.8)=780*0.2=156 | | |
| 每个书架有3层,每层都装50本书,有4个这样的书架,一共装了多少本书? | 答:x=50*3*4=150*4=600 | 1360*5664+8698= | 7703040+8698=7711738 |

Figure 14: Some cases generated by MathGLM on arithmetic tasks and math word problems.

