# OpenReview forum: "GPT Can Solve Mathematical Problems Without a Calculator"
_ICLR.cc/2024/Conference — Submitted to ICLR 2024_

### Official Review · Reviewer_NYtC · 2023-10-27

**Soundness:** 3 good
**Presentation:** 3 good
**Contribution:** 3 good
**Rating:** 8
**Confidence:** 4

**Summary:**

This paper proposes MathGLM, a transformer-based language model specifically designed and trained to excel at mathematical reasoning and arithmetic tasks.

1. MathGLM is trained on a large dataset of arithmetic expressions and sequences, ranging from simple to complex multi-step calculations. This allows it to learn the underlying rules and patterns of arithmetic operations.

2. A step-by-step strategy is used during training, where MathGLM is tasked with generating each intermediate step leading to the final result. This mimics human calculation and helps MathGLM deeply comprehend the calculations.

3. Curriculum learning is used, starting with simpler arithmetic tasks and progressively increasing complexity. This improves efficiency and allows handling of large digit numbers.

4. MathGLM demonstrates significantly higher accuracy on arithmetic tasks compared to GPT-4, ChatGPT and other LLMs. It also achieves comparable performance to GPT-4 on a Chinese math word problem dataset.

**Strengths:**

1. The core idea of specializing a language model for mathematical reasoning is novel and well-motivated. Math is an important domain where current LLMs struggle.

2. The step-by-step training strategy is creative and helps MathGLM learn the intricacies of arithmetic operations. Generating intermediate steps is akin to human math solving.

3. The arithmetic dataset construction process covers various types of math operations and data formats in a principled manner. This diversity is key for strong training.

4. Extensive experiments demonstrate clear performance gains over GPT-4 and other models, validating MathGLM's capabilities. The scaling experiments also provide useful insights.

5. The work is technically sound, clearly presented and easy to follow. The motivation and proposed techniques are intuitive.

**Weaknesses:**

1. While specializing for arithmetic is beneficial, it could compromise more general capabilities. Testing on broader math/reasoning tasks could help characterize tradeoffs.

2. More analysis and examples demonstrating the step-by-step generation process could be useful to understand MathGLM's learned skills.

3. The reasoning behind curriculum learning's benefits is not fully fleshed out. Is it mainly about efficiency gains?

4. How well do the findings transfer to non-Chinese languages? Cross-lingual experiments could help strengthen claims of language-agnostic reasoning.

**Questions:**

Are there any analysis and examples of the errors made by MathGLM? Understanding the remaining limitations could guide future improvements.

For real-world usage, how does MathGLM handle novel word problems outside its training distribution? Experiments on out-of-distribution generalization could be insightful.

---

> ### Author Response · Authors · 2023-11-20
> **Response to  Reviewer NYtC (1/2)**
>
> We sincerely thank you for your valuable comments on our paper. We will explain your concerns point by point.
>
>
> ```
> Weakness 1: While specializing for arithmetic is beneficial, it could compromise more general capabilities. Testing on broader math/reasoning tasks could help characterize tradeoffs.
> ```
>
> **Response**: In our study, we leverage the continue training strategy to address the limitation of fine-tuning that damages the generic ability of the MathGLM.
>
> Here, we report the generic ability of the MathGLM on [Z-bench](https://github.com/zhenbench/z-bench). To evaluate the quality of responses generated by the model, we score responses with the powerful GPT-4 on a scale of 1-10 based on factors such as correctness (high priority), helpfulness, relevance, depth, innovativeness, and level of detail.
>
>
> |Model | Math | Text|
> |----|----|----|
> |GLM-10B | 0% | 5.91 |
> |MathGLM-10B | 56.6% | 5.35 |
>
>
>
> Compared to GLM-10B, MathGLM-10B shows a significant improvement in handling math tasks, with an accuracy of 56.6% compared to GLM's 0%. However, it is also noticeable that there's a slight reduction in the performance on text tasks for MathGLM-10B (5.35) compared to the original GLM (5.91). While MathGLM-10B excels in mathematics, its ability to handle general text tasks is marginally impacted. However, this reduction is not drastic, indicating that the continue training using a hybrid dataset, which includes both mathematical and general text data, helps in retaining the model's overall linguistic competence.
>
> ```
> Weakness 2: More analysis and examples demonstrating the step-by-step generation process could be useful to understand MathGLM's learned skills.
> ```
> **Response**: We have included the impact of the step-by-step generation process in the Appendix, particularly in Section D.6. By leveraging the step-by-step generation process, the accuracy on MathGLM-500M rises from 31.96% to 89.57%, while for MathGLM-2B, it increases from 40.76% to 93.03%. This suggests that the step-by-step process is highly effective in enhancing the model's performance. This could be due to the process allowing the model to break down complex problems into smaller, more manageable steps, thereby reducing errors that can accumulate in a single-step solution.
>
> ```
> Weakness 3: The reasoning behind curriculum learning's benefits is not fully fleshed out. Is it mainly about efficiency gains?
> ```
> **Response**: We leverage curriculum learning to enhance inference performance. This approach involves a gradual introduction of increasingly complex tasks during training. Initially, we focus on simpler training instances, specifically within the 5-digit range. This foundational stage allows the model to efficiently grasp basic calculation rules.
>
> Once the model achieves stable convergence and satisfactory performance on the test dataset, we then implement curriculum learning as a strategic advancement. We introduce a new set of training data that includes numbers ranging from 5 to 12 digits. This methodical escalation in complexity is designed to incrementally challenge and refine the model's capabilities.
>
> Here, we report the performance on MathGLM-2B without using curriculum learning. From the results, it is evident that the use of curriculum learning has a significant impact on the performance of MathGLM-2B. The model, when trained without curriculum learning, achieves an accuracy of 88.70%. However, with the implementation of curriculum learning, the accuracy increases to 93.03%.
>
> |Model | Accuracy |
> |----|----|
> |MathGLM-2B (w/o curriculum learning) | 88.70% |
> |MathGLM-2B | 93.03% |

---

> > ### Author Response · Authors · 2023-11-20
> > **Response to Reviewer NYtC (2/2)**
> >
> > ```
> > Weakness 4: How well do the findings transfer to non-Chinese languages? Cross-lingual experiments could help strengthen claims of language-agnostic reasoning.
> > ```
> > **Response**: In this study, our primary focus is indeed on Chinese language datasets. To conduct cross-lingual experiments, we develop a Chinese-English mathematical model called MathGLM2-6B. Here, we report a preliminary experimental result on our MathGLM2-6B. We evaluate MathGLM2-6B with the same-scale English mathematical models on three classical inference datasets: Math23K, GSM8K, and MATH. The results indicate that MathGLM2-6B demonstrates promising cross-lingual capabilities. On Math23K, a Chinese language dataset, MathGLM2-6B shows competitive performance with a 61.8% accuracy, slightly behind GPT-4's 63.3%. On GSM8K, MathGLM2-6B achieves an accuracy of 63.00%, slightly lower than the similar-scale model MetaMath-7B. In the MATH dataset, MathGLM2-6B achieves an accuracy of 26.76%. In comparison to other  same-scale models, MathGLM2-6B is ahead of WizardMath-7B, Mammonth-7B, and MetaMath-7B.
> >
> >
> > |Model | Math23K | GSM8K| MATH |
> > |----|----|----|----|
> > |GPT-3.5 | 52.4% |80.8% | 34.1% |
> > |GPT-4 | 63.3% | 92% | 42.5% |
> > |WizardMath-7B | - | 54.9% |10.70% |
> > |Mammonth-7B | - | 50.50% |10.40% |
> > |MetaMath-7B | - | 66.50% | 19.80% |
> > |MathGLM2-6B | 61.80% | 63.00% | 26.76% |
> >
> >
> >
> > ```
> > Question 1: Are there any analysis and examples of the errors made by MathGLM? Understanding the remaining limitations could guide future improvements.
> > ```
> > **Response**: In the Appendix of our manuscript, specifically Figure 12, we provide a comprehensive analysis of the errors made by MathGLM-10B in solving math word problems. This analysis identifies various types of errors, including misunderstanding the question, calculation mistakes, incorrect application of mathematical knowledge, and common-sense errors. For instance, in cases of misunderstanding the question, the model sometimes confuses terms like 'more than' with 'less than', leading to wrong solutions. Calculation errors occur when the model makes arithmetic mistakes, often producing partially correct answers but failing to arrive at the correct final solution. Knowledge errors are identified when the model incorrectly applies mathematical concepts, such as formulas for area and volume. Common-sense errors are instances where the model's reasoning deviates from typical human logic, for example, in understanding relationships between velocity and distance.
> >
> > Additionally, we discuss strategies to mitigate these errors in future versions of MathGLM. This involves improving the training data to encompass a wider variety of problem types and incorporating more comprehensive problem-solving procedures. We also consider using a larger model with more training parameters, specifically fine-tuned for math-related tasks. Our ongoing research efforts aim to address these shortcomings, and we expect substantial enhancements in the subsequent iterations of MathGLM.
> >
> > ```
> > Question 2: For real-world usage, how does MathGLM handle novel word problems outside its training distribution? Experiments on out-of-distribution generalization could be insightful.
> > ```
> > **Response**: For real-world usage, we have conducted experiments on the K6 dataset, a collection of problems specifically designed to test out-of-distribution generalization. The K6 dataset, not included in MathGLM’s training data, is collected from the renowned educational institution, TAL AI Lab. The dataset consists of math problems for each grade level, with each grade containing approximately 100 problems. The wide-ranging nature of these math word problems empowers us to gauge the model’s efficacy across an array of difficulty gradients and academic grades.
> >
> > Applying MathGLM to the K6 dataset without additional training allows us to observe its inherent generalization capabilities. As detailed in Figure 6, MathGLM demonstrates notable success in tackling a variety of grade levels compared with the majority of Chinese LLMs. However, we observe that MathGLM's effectiveness in certain grades is not as high as GPT-4's, indicating room for improvement in its performance across the full spectrum of grade levels.

---

> > > ### Comment · Reviewer_NYtC · 2023-11-23
> > > **Official Comment by Reviewer NYtC**
> > >
> > > Thank you for the responses.  They provide clarification on the questions raised.  After discussing with other reviewers, I will reassess my score.

---

### Official Review · Reviewer_mwYm · 2023-10-27

**Soundness:** 1 poor
**Presentation:** 3 good
**Contribution:** 2 fair
**Rating:** 3
**Confidence:** 4

**Summary:**

In this paper, the authors introduce two new datasets that can be used to improve pre-training and fine-tuning of large language models or, more generally, large-scale Transformer models. One dataset contains a large set of arithmetic problems, while the other represents a refined version of Ape210K, which has been augmented with step-by-step solution procedures to solve math word problems involving natural language. The authors exploit these datasets to train a series of Transformer-based language models and show that they indeed achieve more accurate performance in arithmetic and word problem tasks compared to GPT models or other LLMs.

**Strengths:**

-	The article is generally clear and well-written. The research questions are well-motivated.
-	Investigating the arithmetic and mathematical abilities of Transformer-based architectures and LLMs is an important and timely research area.
-	Reconstructing the Ape210K dataset by adding step-by-step solutions constitutes an interesting extension to the available training corpora (which would become even more useful if the dataset would be made publicly available).
-	The authors also analyze the impact of problem difficulty (e.g., grade levels) and the error distribution.

**Weaknesses:**

-	The training/testing setup used in the present work differ from those used in similar work, making it challenging to compare the current results with previous contributions. Overall, it seems that the advantage in the reported benchmarks mostly (only?) stems from the use of an extended training set containing math problems, rather than from architectural innovations. This would still constitute an interesting finding, but it should be demonstrated using out-of-distribution test instances (see next point).
-	The authors claim that MathGLM has a “profound understanding of the complex calculation process” and “effectively learns the underlying rules and principles of arithmetic operations”, however I do not think that its generalization abilities have been properly evaluated.
-	There are a few methodological details than require clarification (see questions below).
-	The paper does not include any Reproducibility Statement or any pointer to source code repositories, which makes it difficult to replicate the simulations and the experimental setup.

**Questions:**

-	The authors say that MathGLM learns to solve arithmetic tasks “by integrating a step-by-step strategy into its architecture”. However, it is not clear how the model architecture actually implements step-by-step reasoning process (from the description, it seems that such feature is just a property of the solution format, rather than of the architecture design). This point should be clarified.
-	In order to properly test for generalization the authors should demonstrate that the model can solve problems outside the training distribution (e.g., involving much longer numbers, and much more operands, see for example https://arxiv.org/abs/2207.02536). At present, an alternative (and more parsimonious) explanation is simply that the larger-scale of the training data allows to the model to memorize a more consistent amount of arithmetic knowledge.
-	The authors say that “To assess the generalization ability of MathGLM beyond the 5-digit range, a set of 50,000 training records involving numbers within the 12-digit range are introduced into the training dataset”. This does not guarantee that generalization is properly assessed; it rather shows that by adding more training samples from the testing range the performance increases, which is expected (also see https://arxiv.org/abs/2306.15400).
-	It is not clear whether the curriculum learning strategy is beneficial since there is no comparison with a non-curriculum counterpart.
-	It is not clear how the Ape210K dataset was reconstructed. Were the step-by-step solutions generated in an automatic way? If so, how was their quality verified?
-	What is the rationale of using different models for the Arithmetic task and the Math Word Problems? Shouldn’t the same MathGLM model be able to solve both types of problems? The authors say that “our goal is to simultaneously advance both mathematical reasoning and arithmetical calculation capabilities of LLMs, addressing both aspects at the same time”, but from my understanding they trained separate models for the Arithmetic and MWP datasets (the “Training Strategy” section at pg. 5 should be expanded and described in a much clearer way).
-	The authors should more carefully explain how GPT models were tested. Which prompting methods were used to probe these models? How did performance change when using more advanced (e.g., Chain-of-though) prompting strategies?
-	The title is misleading, since it suggests that models from the GPT family (e.g., ChatGPT, GPT-4) achieve the best accuracy, while in fact the authors are tuning a model from the GLM family. A better option could be to just use “LLMs” as a more general term?
-	“GLM” has not been properly defined in the introduction (I suggest including both the acronym description and the reference paper).
-	The manuscript content is often redundant; I suggest removing duplicate (or similar) sentences.

---

> ### Author Response · Authors · 2023-11-20
> **Response to Reviewer mwYm (1/3)**
>
> We sincerely thank you for your valuable comments on our paper. We will explain your concerns point by point.
>
> ```
> Weakness 2: The authors claim that MathGLM has a “profound understanding of the complex calculation process” and “effectively learns the underlying rules and principles of arithmetic operations”, however I do not think that its generalization abilities have been properly evaluated.
> ```
> **Response**: Please refer to the Response in Question 2.
>
> ```
> Weakness 4: The paper does not include any Reproducibility Statement or any pointer to source code repositories, which makes it difficult to replicate the simulations and the experimental setup.
> ```
> **Response**: Here, we provide an anonymous repository (https://anonymous.4open.science/r/MathGLM-anonymous-3085) to release our datasets and code.
>
>
>
> ```
> Question 1: The authors say that MathGLM learns to solve arithmetic tasks “by integrating a step-by-step strategy into its architecture”. However, it is not clear how the model architecture actually implements step-by-step reasoning process (from the description, it seems that such feature is just a property of the solution format, rather than of the architecture design). This point should be clarified.
> ```
> **Response**: We apologize for the confusion. In this study, we do not design a new architecture to solve arithmetic tasks. Instead, we construct a pre-training dataset that includes a variety of arithmetic expressions, encompassing different types of arithmetic operations and various types of numbers.
>
> The key to facilitating the step-by-step reasoning is in the way these expressions are structured and presented in the dataset. This structure of a pre-training dataset provides the model with examples of grasping calculation rules during training and inference.
>
> Furthermore, to enhance the model's ability to handle arithmetic tasks, we employ a unique tokenization strategy. Each digit in an arithmetic expression is tokenized as a distinct token. This approach allows MathGLM to focus on each component of the problem individually, facilitating more accurate computations and enabling the model to follow the step-by-step reasoning pattern inherent in the dataset.
>
> By training MathGLM on this carefully constructed dataset with its unique tokenization method, the model implicitly learns a step-by-step reasoning process, despite not having an architecture explicitly designed for this purpose. Compared with the method that directly calculates the answer of the question, the step-by-step methodology enhances the model's accuracy in arithmetic tasks (Cf. Figure 10).
>
>
>
>
>
> ```
> Question 2: In order to properly test for generalization the authors should demonstrate that the model can solve problems outside the training distribution (e.g., involving much longer numbers, and much more operands, see for example https://arxiv.org/abs/2207.02536). At present, an alternative (and more parsimonious) explanation is simply that the larger-scale of the training data allows to the model to memorize a more consistent amount of arithmetic knowledge.
> ```
> **Response**: We thank the reviewer for highlighting the importance of testing for generalization beyond the training distribution. To test the generalizability of MathGLM,  we evaluate the generalizability ability of MathGLM on arithmetic problems that involve numbers beyond the 12-digit range.
>
>
> |Model | 13-D | 14-D| 15-D |
> |----|----|----|----|
> |MathGLM-500M | 16.10% | 10.31% | 18.07% |
> |MathGLM-2B | 28.74% | 36.08% | 37.34% |
>
>
>
>
>
>
>
> ```
> Question 3: The authors say that “To assess the generalization ability of MathGLM beyond the 5-digit range, a set of 50,000 training records involving numbers within the 12-digit range are introduced into the training dataset”. This does not guarantee that generalization is properly assessed; it rather shows that by adding more training samples from the testing range the performance increases, which is expected (also see https://arxiv.org/abs/2306.15400).
> ```
> **Response**: We appreciate the reviewer’s critical observation regarding our methodology for assessing the generalization ability of MathGLM. In our evaluation setting, our aim is to demonstrate MathGLM's enhanced performance in generalization with a notably smaller dataset. This dataset is significantly smaller in comparison to the volume of data used during the initial training phase of MathGLM. This approach underscores the efficiency and effectiveness of MathGLM in learning and adapting to new data.
>
>
> To rigorously test the model's inherent generalization capabilities, we evaluate the generalizability ability of MathGLM on arithmetic problems that involve numbers beyond the 12-digit range. The detailed results can be found in Response to Question 3.

---

> ### Author Response · Authors · 2023-11-20
> **Response to Reviewer mwYm (2/3)**
>
> ```
> Question 4: It is not clear whether the curriculum learning strategy is beneficial since there is no comparison with a non-curriculum counterpart.
> ```
> **Response**: We leverage curriculum learning to enhance inference performance and increase model efficiency. This approach involves a gradual introduction of increasingly complex tasks during training. Initially, we focus on simpler training instances, specifically within the 5-digit range. This foundational stage allows the model to efficiently grasp basic calculation rules.
>
> Once the model achieves stable convergence and satisfactory performance on the test dataset, we then implement curriculum learning as a strategic advancement. We introduce a new set of training data that includes numbers ranging from 5 to 12 digits. This methodical escalation in complexity is designed to incrementally challenge and refine the model's capabilities.
>
> Here, we report the result of the performance on MathGLM-2B without using curriculum learning. From the results, it is evident that the use of curriculum learning has a significant impact on the performance of MathGLM-2B. The model, when trained without curriculum learning, achieves an accuracy of 88.70%. However, with the implementation of curriculum learning, the accuracy increases to 93.03%.
>
> |Model | Accuracy |
> |----|----|
> |MathGLM-2B (w/o curriculum learning) | 88.70% |
> |MathGLM-2B | 93.03% |
>
>
>
>
> ```
> Question 5: It is not clear how the Ape210K dataset was reconstructed. Were the step-by-step solutions generated in an automatic way? If so, how was their quality verified?
> ```
> **Response**: In our reconstruction of the Ape210K dataset, we employ a Python script to enable a detailed, step-by-step calculation for each math problem (Cf. Figure 3). This enhancement in the dataset is specifically designed to bolster MathGLM's proficiency in grasping the intricate calculation rules essential for solving math word problems. By navigating through these step-wise computations, MathGLM is trained to not only reach the correct answer but also to comprehend the underlying mathematical reasoning. Furthermore, we plan to make the enhanced Ape210K dataset publicly available for open research, contributing to the broader academic community.
>
>
> ```
> Question 6 : What is the rationale of using different models for the Arithmetic task and the Math Word Problems? Shouldn’t the same MathGLM model be able to solve both types of problems? The authors say that “our goal is to simultaneously advance both mathematical reasoning and arithmetical calculation capabilities of LLMs, addressing both aspects at the same time”, but from my understanding they trained separate models for the Arithmetic and MWP datasets (the “Training Strategy” section at pg. 5 should be expanded and described in a much clearer way).
> ```
> **Response**: The reason why we use different models for arithmetic tasks and mathematical word problems (MWP) in our study is that the motivations we study are different.
>
> For arithmetic tasks, we aim to address the misconception about the capabilities of large language models (LLMs) in performing arithmetic operations. MathGLM challenges the prevailing notion about LLMs' limitations in arithmetic operations, particularly with longer digits and more complex calculations like decimals and fractions. The utilized step-by-step strategy can significantly improve the arithmetic performance (Cf. Figure 10).
>
> Conversely, for MWPs, we recognize the limitations of the Ape210K dataset, which originally provided direct answers without detailed computations. To address this, we reconstruct the Ape210K dataset using a step-by-step solution strategy and fine-tune GLM models on this reconstructed dataset, specifically for MWPs.
>
> We apologize for the confusion and will revise the statements in the "Training Strategy" section.
>
>
>
> ```
> Question 7: The authors should more carefully explain how GPT models were tested. Which prompting methods were used to probe these models? How did performance change when using more advanced (e.g., Chain-of-though) prompting strategies?
> ```
> **Response**: In our evaluation, we use the standard direct prompting method. In this approach, the models are presented with straightforward questions without additional contextual or guiding prompts. This method was chosen to ensure a fair and consistent baseline for comparing the performance of different models.

---

> > ### Author Response · Authors · 2023-11-20
> > **Response to Reviewer mwYm (3/3)**
> >
> > ```
> > Question 8: The title is misleading, since it suggests that models from the GPT family (e.g., ChatGPT, GPT-4) achieve the best accuracy, while in fact the authors are tuning a model from the GLM family. A better option could be to just use “LLMs” as a more general term?
> > ```
> > **Response**: Thanks for your nice suggestions!
> >
> >
> > ```
> > Question 9: “GLM” has not been properly defined in the introduction (I suggest including both the acronym description and the reference paper).
> >
> > ```
> > **Response**: Thanks for your nice suggestions! We will add the description of the GLM model in the Introduction Section in the next version. We have introduced GLM models in Appendix B.2.
> >
> >
> >
> > ```
> > Question 10: The manuscript content is often redundant; I suggest removing duplicate (or similar) sentences.
> > ```
> > **Response**: Thanks for your nice suggestions. We will revise our statements in the next version.

---

> > > ### Comment · Reviewer_mwYm · 2023-11-21
> > > **The revised paper improves on several weak points, however some critical issues remain.**
> > >
> > > I thank the authors for their thoughtful responses. I think that several of my concerns would indeed be fixed by implementing the actions mentioned by the authors (e.g., providing an open repository, improving the methodological description and the discussion, etc.).
> > >
> > > At the same time, I think that the paper has some critical weaknesses that would not be addressed according to the planned revision. In particular, I am not convinced that the current results support the idea that the model has a "profound understanding of the complex calculation process", neither that it "effectively learns the underlying rules and principles of arithmetic operations". The new results related to operations with more digits do not add evidence about this weaknesses and only partially explore the issue of out-of-distribution generalization (e.g., what happens when the problem also involves more operands, or when the structure of the computation [e.g., nestedness] significantly changes w.r.t. the training distribution?). Furthermore, I think that the comparison with GPT models should involve the exploration of a few advanced prompting strategies, since we know that these LLMs achieve much higher accuracy when properly prompted. The authors argue that they did not use advanced prompting "to ensure a fair and consistent baseline", but since their MathGLM model has been trained on a different dataset (containing step-by-step problem solutions) I wonder why we shouldn't encourage the LLMs to similarly exploit step-by-step reasoning for solving the task.
> > >
> > > Overall, I find this paper of some interest and I think that its quality could improve after the planned revision, but in my opinion it still contains critical flaws that would not justify publication in a top-tier venue like ICLR.

---

> > > > ### Author Response · Authors · 2023-11-22
> > > > **Response to Reviewer mwYm -- some critical issues**
> > > >
> > > > We deeply appreciate your continued feedback and recognition of our efforts to improve the manuscript. Your insights are invaluable, and we have meticulously considered each point to enhance the quality of our work. Below, we address the critical weaknesses you highlighted.
> > > >
> > > > ```
> > > > Concern 1: The new results related to operations with more digits do not add evidence about this weaknesses and only partially explore the issue of out-of-distribution generalization (e.g., what happens when the problem also involves more operands, or when the structure of the computation [e.g., nestedness] significantly changes w.r.t. the training distribution?).
> > > > ```
> > > > **Response**: Thanks for your valuable suggestions on evaluating out-of-distribution generalization.
> > > >
> > > >
> > > > Here, we expand the scope of our experiments to evaluate the generalization ability of MathGLM. In our study, the pre-training dataset for arithmetic tasks is designed to encompass a wide spectrum of arithmetic operations, spanning from straightforward 1-atomic operation to more complex 9-atomic operations. To evaluate the generalization ability of MathGLM, we conduct additional evaluations with different atomic operations ranging from 10-atomic to 15-atomic involving numbers within the 12-digit range.
> > > >
> > > > **Different atomic operations within the 12-digit range:**
> > > > |Model | 10-atomic | 11-atomic| 12-atomic |  13-atomic| 14-atomic | 15-atomic |
> > > > |----|----|----|----|----|----|----|
> > > > |MathGLM-2B | 52.05%| 46.48% | 47.83% | 54.17% | 39.06% | 57.41% |

---

> > > > > ### Author Response · Authors · 2023-11-22
> > > > > **Response to Reviewer mwYm -- some critical issues**
> > > > >
> > > > > ```
> > > > > Concern 2: Furthermore, I think that the comparison with GPT models should involve the exploration of a few advanced prompting strategies, since we know that these LLMs achieve much higher accuracy when properly prompted.  The authors argue that they did not use advanced prompting "to ensure a fair and consistent baseline", but since their MathGLM model has been trained on a different dataset (containing step-by-step problem solutions) I wonder why we shouldn't encourage the LLMs to similarly exploit step-by-step reasoning for solving the task.
> > > > > ```
> > > > > **Response**: We appreciate the opportunity to clarify certain aspects of our evaluation methodology for GPT models.
> > > > >
> > > > > Firstly, it's crucial to note that in our study, **GPT models (like GPT-3.5 and GPT-4) utilize a formula similar to the Chain of Thought (CoT) approach for generating answers.** This process is not in one step; it involves producing solutions in a step-by-step manner until the final answer is reached. This method aligns with the CoT strategy, and it is a key element in our evaluation.
> > > > >
> > > > > To address your concern, we incorporate an advanced prompting technique, similar to CoT, in our assessment of GPT models. The CoT prompt is represented as "Let's calculate the following arithmetic expression step by step.". The results for arithmetic tasks are reported as follows:
> > > > >
> > > > > **For arithmetic tasks:**
> > > > > |Model | Accuracy |
> > > > > |----|----|
> > > > > |GPT-4-CoT|  19.52% |
> > > > > |GPT-4|  18.84% |
> > > > > |ChatGPT-CoT | 10.91% |
> > > > > |ChatGPT | 10.00% |
> > > > > |MathGLM-10M | 61.21% |
> > > > > |MathGLM-100M | 70.28% |
> > > > > |MathGLM-500M | 89.57% |
> > > > > |MathGLM-2B | 93.03% |
> > > > >
> > > > > Here, we provide some examples generated by GPT-4 **without the CoT prompting method**.
> > > > >
> > > > > **Example 1:** Let us calculate the following arithmetic expression: (86/92)+(161/92)=
> > > > > ```
> > > > > To calculate the expression (86/92) + (161/92), we need to add the numerators and keep the common denominator.
> > > > >
> > > > > (86 + 161) / 92 = 247 / 92
> > > > >
> > > > > The fraction cannot be simplified further, so the answer is 247/92.
> > > > > ```
> > > > >
> > > > > **Example 2:**  Let us calculate the following arithmetic expression: (-6971*-3209)/-8739.84--8075*-7708.4355/-9016*8031.75=
> > > > > ```
> > > > > The result of the arithmetic expression is:
> > > > >
> > > > > (-6971 * -3209) / -8739.84 - (-8075 * -7708.4355) / -9016 * 8031.75
> > > > >
> > > > > = 22306539 / -8739.84 - 62336068.5636 / -9016 * 8031.75
> > > > >
> > > > > = -2490.131 + 6900.87263314 * 8031.75
> > > > >
> > > > > = -2490.131 + 55452181.3857175
> > > > >
> > > > > = 55449691.2547175
> > > > > ```
> > > > >
> > > > > Here, we provide some examples generated by GPT-4 **with the CoT prompting method**.
> > > > >
> > > > > **Example 1:** Let us calculate the following arithmetic expression step by step: (86/92)+(161/92)=
> > > > > ```
> > > > > To calculate the arithmetic expression (86/92) + (161/92), we will follow these steps:
> > > > >
> > > > > Step 1: Find the least common denominator (LCD) of the two fractions, which is 92.
> > > > >
> > > > > Step 2: Rewrite the fractions with the LCD as the denominator:
> > > > >
> > > > > (86/92) + (161/92)
> > > > >
> > > > > Step 3: Add the numerators together and keep the denominator the same:
> > > > >
> > > > > 86 + 161 = 247
> > > > >
> > > > > So the expression becomes:
> > > > >
> > > > > 247/92
> > > > >
> > > > > Step 4: Simplify the fraction, if possible:
> > > > >
> > > > > 247/92 = 2.6848 (rounded to four decimal places)
> > > > >
> > > > > Therefore, (86/92) + (161/92) = 2.6848 (rounded to four decimal places).
> > > > > ```
> > > > >
> > > > > **Example 2:** Let us calculate the following arithmetic expression step by step: (-6971*-3209)/-8739.84--8075*-7708.4355/-9016*8031.75=
> > > > > ```
> > > > > To calculate the given arithmetic expression step by step, we can break it down into smaller parts:
> > > > >
> > > > > 1. (-6971 * -3209) / -8739.84 = 22303239.39 / -8739.84 = -2552.0504
> > > > >
> > > > > 2. -8075 * -7708.4355 = 62228903.1132
> > > > >
> > > > > 3. -62228903.1132 / -9016 = 6900.7981
> > > > >
> > > > > 4. 6900.7981 * 8031.75 = 55428370.3038
> > > > >
> > > > > Therefore, (-6971 * -3209) / -8739.84 - (-8075 * -7708.4355) / -9016 * 8031.75 = -55428370.3038.
> > > > > ```

---

> > > > > > ### Author Response · Authors · 2023-11-22
> > > > > > **Response to Reviewer mwYm -- some critical issues**
> > > > > >
> > > > > > Here, we utilize the CoT prompting method to assess GPT models for math word problem (MWP). The CoT prompt is represented as "Let's think step by step.". The results for MWP tasks are reported as follows:
> > > > > >
> > > > > >
> > > > > >
> > > > > >
> > > > > >
> > > > > > **For math word problems:**
> > > > > > |Model | Accuracy |
> > > > > > |----|----|
> > > > > > |GPT-4-CoT| 59.90% |
> > > > > > |GPT-4| 59.57% |
> > > > > > |ChatGPT-CoT | 40.48% |
> > > > > > |ChatGPT| 39.78% |
> > > > > > |MathGLM-10B | 58.68% |
> > > > > >
> > > > > >
> > > > > > Here, we provide some examples generated by GPT-4 **without the CoT prompting method**.
> > > > > >
> > > > > >
> > > > > > **Example 1:** Let us calculate the following math word problem: Wang Yan’s family bought a washing machine and a refrigerator. They spent a total of 6,000 yuan. The price of the refrigerator is the same as that of the washing machine (3/5). Please tell me the price of the washing machine. (王艳家买了一台洗衣机和一台电冰箱，一共花了6000元，电冰箱的价钱是洗衣机的(3/5)，求洗衣机的价钱.)
> > > > > > ```
> > > > > > Let's denote the price of the washing machine as X and the price of the refrigerator as 3/5*X.
> > > > > >
> > > > > > According to the problem, we know that the sum of the price of the washing machine and the refrigerator equals 6000 yuan, therefore:
> > > > > >
> > > > > > X + 3/5*X = 6000
> > > > > >
> > > > > > Multiply everything by 5 to make the equation simpler:
> > > > > >
> > > > > > 5X + 3X = 30000
> > > > > >
> > > > > > Then combine like terms:
> > > > > >
> > > > > > 8X = 30000
> > > > > >
> > > > > > Finally, solving the equation gives us the price of the washing machine:
> > > > > >
> > > > > > X = 30000 / 8 = 3750
> > > > > >
> > > > > > Therefore, the price of the washing machine is 3750 yuan.
> > > > > > ```
> > > > > >
> > > > > >
> > > > > > **Example 2:** Let us calculate the following math word problem: There are 5 baskets of apples with the same weight. If 10kg is taken out from each basket, then the remaining apples are equivalent to the weight of the original 3 baskets. How many kilograms does each basket of apples weigh? (有5筐苹果的重量相等，如果从每筐中取出10kg，那么剩下的苹果相当于原来3筐的重量，原来每筐苹果重多少千克？)
> > > > > > ```
> > > > > > The problem says that there are five buckets of apples with the same weight. If we remove 10kg from each, the total weight will equate to that of three original buckets. From this, we can derive an equation.
> > > > > >
> > > > > > Let's say that the weight of the apples in each bucket is x kg.
> > > > > >
> > > > > > So, the total weight of apples in all five buckets is 5x.
> > > > > >
> > > > > > After removing 10kg from each bucket, the total weight becomes 5x - 5*10 = 5x - 50.
> > > > > >
> > > > > > According to the problem, 5x - 50 = 3x.
> > > > > >
> > > > > > Obtaining x from this equation, we have:
> > > > > >
> > > > > > 5x - 3x = 50,
> > > > > > 2x = 50,
> > > > > > Therefore, x = 50/2 = 25.
> > > > > >
> > > > > > So, the original weight of the apples in each bucket is 25kg.
> > > > > > ```
> > > > > >
> > > > > > Here, we provide some examples generated by GPT-4 **with the CoT prompting method**.
> > > > > >
> > > > > >
> > > > > > **Example 1:** Let us calculate the following math word problem step by step: Wang Yan’s family bought a washing machine and a refrigerator. They spent a total of 6,000 yuan. The price of the refrigerator is the same as that of the washing machine (3/5). Please tell me the price of the washing machine. (王艳家买了一台洗衣机和一台电冰箱，一共花了6000元，电冰箱的价钱是洗衣机的(3/5)，求洗衣机的价钱.)
> > > > > > ```
> > > > > > According to the problem, let's assume that the price of the washing machine is X RMB. Therefore, the price of the refrigerator is (3/5)*X RMB.
> > > > > >
> > > > > > According to the problem, the total cost of the washing machine and refrigerator is 6000 RMB. Therefore we can setup the following equation:
> > > > > >
> > > > > > X + (3/5)*X = 6000
> > > > > >
> > > > > > This simplifies to:
> > > > > >
> > > > > > (8/5)*X = 6000
> > > > > >
> > > > > > Solving for X gives the price of the washing machine:
> > > > > >
> > > > > > X = 6000 * (5/8) = 3750 RMB
> > > > > >
> > > > > > So, the price of the washing machine is 3750 RMB.
> > > > > >
> > > > > > ```
> > > > > >
> > > > > >
> > > > > >
> > > > > > **Example 2:** Let us calculate the following math word problem step by step:  There are 5 baskets of apples with the same weight. If 10kg is taken out from each basket, then the remaining apples are equivalent to the weight of the original 3 baskets. How many kilograms does each basket of apples weigh? (有5筐苹果的重量相等，如果从每筐中取出10kg，那么剩下的苹果相当于原来3筐的重量，原来每筐苹果重多少千克？)
> > > > > > ```
> > > > > > The problem states that there are 5 baskets of apples with equal weight. If you remove 10 kg from each basket, the remaining apples weigh as much as 3 full baskets originally did. To find out the original weight of one basket, we can break down this problem into two main steps.
> > > > > >
> > > > > > Step 1: Calculate the total weight of the apples after removing 10 kg from each basket.
> > > > > > Since the remaining apples weigh as much as 3 full baskets, we know that the total weight after removal is the weight of 3 full baskets.
> > > > > >
> > > > > > Step 2: Calculate the weight of one full basket before the removal.
> > > > > > We know that 5 baskets of apples were reduced by 10 kg. This means that the total reduction is 5 * 10 = 50 kg. To find the total original weight, we therefore add this reduction (50 kg) onto the total weight after removal (the weight of 3 full baskets).
> > > > > >
> > > > > > Let's denote the weight of one full basket as 'x'. Then, the equation we can generate from the problem is:
> > > > > >
> > > > > > 3x = 5x - 50.
> > > > > >
> > > > > > Solving this equation for x will give us the weight of a full basket before the apples were removed.
> > > > > >
> > > > > > Step 3: Solve the equation
> > > > > > By moving the terms around, we get:
> > > > > > 5x - 3x = 50,
> > > > > > which simplifies to:
> > > > > > 2x = 50.
> > > > > >
> > > > > > We then solve for x by dividing both sides by 2:
> > > > > > x = 50 / 2 = 25.
> > > > > >
> > > > > > So the original weight of each basket of apples was 25 kg.
> > > > > > ```

---

### Official Review · Reviewer_Bh6z · 2023-11-01

**Soundness:** 4 excellent
**Presentation:** 4 excellent
**Contribution:** 3 good
**Rating:** 8
**Confidence:** 3

**Summary:**

The paper has a full study of LLM on math problems, with the focus of multi-digit complex operations and math problems in regular text. The dataset are created, and the LLM of different sizes are fine tuned. The MathGLM has been evaluated on many setup and ablation. The new model has proven better performance than the GPT-4 on the two goals.

**Strengths:**

The math accuracy belongs to one of the core challenge of LLM. The paper has very good CoT dataset and gets enhanced performance compared to the GPT-4 model. The paper appears rather useful among many scholars from relevant area.

**Weaknesses:**

Could we extend the evaluation of the new model and see the performance on non-math tasks? The math focus finetune may have reduced the performance on other tasks, and it is good to know how good / bad that would be.

**Questions:**

Maybe a followup work would seem how to train model for middle-school level math problems, how the size of dataset / model would scale for that

---

> ### Author Response · Authors · 2023-11-20
> **Response to Reviewer Bh6z**
>
> Thank you for your insightful and well-articulated points. We will explain your concerns point by point.
>
>
>
> ```
> Weakness 1: Could we extend the evaluation of the new model and see the performance on non-math tasks? The math focus finetune may have reduced the performance on other tasks, and it is good to know how good / bad that would be.
> ```
>
> **Response**: In fact, fine-tuning the model specifically for mathematical tasks may have implications for its performance in other domains. This specialized fine-tuning could lead to a reduction in the model's effectiveness in handling non-mathematical tasks.
>
> In order to reduce this effect, we leverage a hybrid dataset dataset that includes both mathematical and text data for the continue training strategy. Specifically, we use a public Chinese text data [Chinese-Vicuna](https://huggingface.co/datasets/Chinese-Vicuna/instruct_chat_50k.jsonl), a rich source of public Chinese text dataset, with our reconstructed Ape210K dataset for continue training. This approach aims to maintain the model's proficiency in general language tasks while enhancing its mathematical abilities.
>
>
> To evaluate the performance of the model on non-math tasks, we use the [Z-bench](https://github.com/zhenbench/z-bench), consisting of 63 diverse questions. To evaluate the quality of responses generated by the model, we score responses with the powerful GPT-4 on a scale of 1-10 based on factors such as correctness (high priority), helpfulness, relevance, depth, innovativeness, and level of detail. The results are as follows:
>
>
> |Model | Math | Text|
> |----|----|----|
> |GLM-10B | 0% | 5.91 |
> |MathGLM-10B | 56.6% | 5.35 |
>
>
>
> MathGLM-10B demonstrates a significant improvement in the mathematical task, achieving a 56.6% accuracy, a significant leap from GLM's 0%. However, it's important to note the slight decrease in performance on general text tasks for MathGLM-10B (5.35) compared to GLM-10B (5.91). While MathGLM-10B excels in mathematics, its ability to handle general text tasks is marginally impacted.
>
>
> Nevertheless, the marginal reduction in text task performance indicates that the continue training strategy of using a hybrid training dataset has been effective in preserving the model's overall linguistic abilities. Therefore, while there is a noticeable specialization in mathematical tasks, MathGLM-10B maintains a competent level of performance in general language processing.
>
>
>
> ```
> Question 1: Maybe a followup work would seem how to train model for middle-school level math problems, how the size of dataset / model would scale for that
> ```
>
> **Response**: Thank you for the suggestion regarding follow-up work. We are indeed in the process of developing MathGLM-v2, a specialized model designed specifically for middle-school level mathematics.
>
> The pre-training dataset with MathGLM-v2 encompasses a wide spectrum of mathematical problems relevant to middle-school education. This dataset is not just extensive in size but also diverse in content, covering areas from basic arithmetic to more complex algebraic concepts. As for the model itself, MathGLM-v2 is trained on a larger model with 12B training parameters.
>
>
> Preliminary results are promising, showcasing the model's ability to grasp fundamental mathematical concepts and apply them in problem-solving. As we progress, we aim to refine MathGLM-v2 further, enhancing its accuracy and ability to generalize across a wider array of math problems.
>
>
> We are working towards releasing MathGLM-v2 in the near future.

---

### Official Review · Reviewer_GLqj · 2023-11-01

**Soundness:** 2 fair
**Presentation:** 3 good
**Contribution:** 2 fair
**Rating:** 5
**Confidence:** 4

**Summary:**

The authors propose using LLMs to perform complex mathematical computations. To prove this theory, they trained a model called MathGLM on a dataset with multi-step arithmetic operations and math problems described in text. They verify their results on the APE test set, as well as a K6 dataset they proposed, which consists of elementary-school math word problems. They demonstrate that on a constructed dataset of complex mathematical computations, their model outperforms GPT-4.

**Strengths:**

1. Interesting perspective on using LLMs to conduct complex mathematical computations without the use of tools.
2. The paper states its theory and results clearly.

**Weaknesses:**

1. The claim regarding motivation is not robust. While MathGLM achieves a high accuracy of 93.03% on the constructed dataset for complex computations, these calculations can be done with 100% accuracy using other tools.
2. The computation is limited to addition, subtraction, multiplication, division, and exponentiation. The method probably wouldn't generalize well to more intricate computations such as log, sin, etc. Moreover, mathematics should aim for complete accuracy, so utilizing LLMs for these calculations isn't a suitable strategy, especially considering the costly pretraining involved for computations that other tools can resolve more efficiently. Instead, LLMs should concentrate on providing more insight and higher-level strategies for solving math problems.
3. The primary math word problem datasets are APE and K12, both of which are in Chinese. There were no experiments conducted on popular math datasets like MATH and GSM8K. Since GPT-4 is primarily trained in English, and MathGLM is fine-tuned for Chinese math word problems, such a comparison might not be valid. The advantage of MathGLM could be due to the language, rather than its proficiency in resolving math problems.

**Questions:**

See Weaknesses and,

This paper uses LLMs to conduct complex mathematical computations. This is a novel approach, but the motivation is weak because using LLMs for complex mathematical computations lacks accuracy and generalizability. Additionally, there is a lack of experimentation on MATH and GSM8K, as the primary comparisons are made using Chinese mathematical datasets.

Correctness: 3: Some of the paper’s claims have minor issues. A few statements are not well-supported, or require small changes to be made correct.

Technical Novelty And Significance: 2: The contributions are only marginally significant or novel.

Empirical Novelty And Significance: 2: The contributions are only marginally significant or novel.

---

> ### Author Response · Authors · 2023-11-20
> **Response to Reviewer GLqj (1/2)**
>
> Thanks for your appreciation of our work and valuable suggestions! We will explain your concerns point by point.
>
> ```
> Weakness 1: The claim regarding motivation is not robust. While MathGLM achieves a high accuracy of 93.03% on the constructed dataset for complex computations, these calculations can be done with 100% accuracy using other tools.
> ```
> **Response**: The motivation of our paper is to address the misconception about the capabilities of large language models (LLMs) in performing arithmetic operations.
>
> 1. ***Redefining LLMs Capabilities on Arithmetic Tasks***
> MathGLM challenges the prevailing notion about LLMs' limitations in arithmetic operations, particularly with longer digits and more complex calculations like decimals and fractions. This contributes to the field by expanding the understanding and capabilities of LLMs in mathematical computations.
>
> 2. ***Beyond Mere Calculations***
> We acknowledge that other tools may achieve 100% accuracy. However, we clarify that the uniqueness of MathGLM lies in its ability to handle complex arithmetic operations within the framework of large language models. This allows it to not only perform calculations but also understand and process complex math word problems, which is beyond the capability of many conventional tools.
>
> 3. ***Enhanced Performance in Math Word Problems***
> For math word problems, we incorporate this step-by-step strategy into the original Ape210K dataset, which significantly bolsters MathGLM's answer accuracy (42.29% performance gains).
>
>
> ```
> Weakness 2: The computation is limited to addition, subtraction, multiplication, division, and exponentiation. The method probably wouldn't generalize well to more intricate computations such as log, sin, etc. Moreover, mathematics should aim for complete accuracy, so utilizing LLMs for these calculations isn't a suitable strategy, especially considering the costly pretraining involved for computations that other tools can resolve more efficiently. Instead, LLMs should concentrate on providing more insight and higher-level strategies for solving math problems.
> ```
> **Response**: In this study, we focus on addressing the misconception about the capabilities of large language models (LLMs) in performing arithmetic operations. MathGLM is currently limited to basic operations like addition, subtraction, multiplication, division, and exponentiation.
>
> It's important to note that functions like logarithm and sine, which are more complex, are not directly computable by humans without tools. We often rely on calculators for these function calculations, suggesting that LLMs may not be critically needed for such intricate calculations.
>
>
> We agree with your claim that LLMs should concentrate on providing more insight and higher-level strategies for solving math problems. In our study, we find that calculation errors are a significant error type in solving math word problems (Cf. Figure 9). In order to enhance the performance of MathGLM-10B in solving math word problems, we incorporate an additional 5,500 calculation-specific training data into the reconstructed Ape210K, and evaluate performance on the Ape210K test dataset. From the results, we can observe that by adding calculation-specific data, MathGLM-10B's performance is significantly improved compared with the original MathGLM-10B (reported in our manuscript). Therefore, the study of calculation also plays an important role in solving math word problems.
>
>
>
> |Model | Accuracy |
> |----|----|
> |GPT-3.5 | 39.78% |
> |GPT-4 | 59.57% |
> |MathGLM-10B | 58.68% |
> |MathGLM-10B (with calculation-specific data)  | 63.20% |

---

> ### Author Response · Authors · 2023-11-20
> **Response to Reviewer GLqj (2/2)**
>
> ```
> Weakness 3: The primary math word problem datasets are APE and K12, both of which are in Chinese. There were no experiments conducted on popular math datasets like MATH and GSM8K. Since GPT-4 is primarily trained in English, and MathGLM is fine-tuned for Chinese math word problems, such a comparison might not be valid. The advantage of MathGLM could be due to the language, rather than its proficiency in resolving math problems.
> ```
> **Response**: In this study, our primary focus is indeed on Chinese language datasets. However, it's important to clarify that the effectiveness of MathGLM is not solely attributable to the language aspect.
>
>
> To further verify our claim, we develop a Chinese-English mathematical model called MathGLM2-6B. Here, we report a preliminary experimental result on our MathGLM2-6B. We evaluate MathGLM2-6B with the same-scale English mathematical models on three classical inference datasets: Math23K, GSM8K, and MATH. The results indicate that MathGLM2-6B demonstrates promising cross-lingual capabilities. On Math23K, a Chinese language dataset, MathGLM2-6B shows competitive performance with a 61.8% accuracy, slightly behind GPT-4's 63.3%. On GSM8K, MathGLM2-6B achieves an accuracy of 63.00%, slightly lower than the similar-scale model MetaMath-7B. In the MATH dataset, MathGLM2-6B achieves an accuracy of 26.76%. In comparison to other same-scale models, MathGLM2-6B is ahead of WizardMath-7B, Mammonth-7B, and MetaMath-7B.
>
>
> In conclusion, while language is an important aspect of our model, the primary advantage of MathGLM lies in its ability to understand and solve complex mathematical problems.
>
>
>
> |Model | Math23K | GSM8K| MATH |
> |----|----|----|----|
> |GPT-3.5 | 52.4% |80.8% | 34.1% |
> |GPT-4 | 63.3% | 92% | 42.5% |
> |WizardMath-7B | - | 54.9% |10.70% |
> |Mammonth-7B | - | 50.50% |10.40% |
> |MetaMath-7B | - | 66.50% | 19.80% |
> |MathGLM2-6B | 61.80% | 63.00% | 26.76% |
>
>
>
>
> ```
> Question 1: This paper uses LLMs to conduct complex mathematical computations. This is a novel approach, but the motivation is weak because using LLMs for complex mathematical computations lacks accuracy and generalizability. Additionally, there is a lack of experimentation on MATH and GSM8K, as the primary comparisons are made using Chinese mathematical datasets.
> ```
> **Response**: The motivation of our paper can refer to the response in Weakness 1 and the experimentation on MATH and GSM8K can refer to Weakness 3.

---

### Meta-Review · Area_Chair_4ZCs · 2023-12-21

**Metareview:**

After a careful consideration of the reviewers' comments, rebuttals, and the authors' response, the decision is to reject the paper from this conference cycle. While the authors have put considerable effort into addressing some of the concerns raised by the reviewers, there remain critical issues that significantly impact the paper's suitability for acceptance at this time.

The reviewers commend the authors for a well-written paper and appreciate the novel approach to enhancing the mathematical reasoning capabilities of language models. The step-by-step training strategy and curriculum learning aspects are recognized as creative and potentially impactful contributions to the field. However, there are several areas where the paper falls short:

1. Generalization Ability: Despite additional experiments, the reviewers are not convinced that MathGLM exhibits a "profound understanding" or "effective learning" of arithmetic operations beyond the training distribution. The experiments do not fully explore out-of-distribution generalization, such as problems with more operands or varying computational structures, which are crucial for substantiating the claims of understanding and generalization.

2. Comparison to GPT Models: The paper's comparison with GPT models is found to be lacking. While advanced prompting strategies (such as Chain of Thought) were eventually considered in the response to reviewers, it is unclear why these were not part of the initial evaluation, given that the MathGLM model was trained on step-by-step problem solutions. A fair comparison would require leveraging the full potential of prompting strategies for both models.

3. Methodological Clarity: Certain methodological details, including the implementation of curriculum learning and the quality verification of step-by-step solutions in the reconstructed datasets, are not clearly described. This lack of clarity raises concerns about the replicability and robustness of the results.

4. Cross-Lingual Experiments: The authors have made attempts to address the concern about language-specific advantages by introducing cross-lingual experiments. However, the reviewers note that these experiments are preliminary and do not sufficiently strengthen the claims of language-agnostic reasoning capabilities.

5. Trade-offs in Specialization: The paper does not adequately address the potential trade-offs that specializing a model for arithmetic may have on its general capabilities. It is essential to understand how MathGLM performs on a broader range of reasoning tasks to evaluate its practical utility fully.

In light of these issues, the paper does not meet the acceptance criteria for this conference. The authors are encouraged to address the above concerns and consider resubmitting their work in the future.

**Justification For Why Not Higher Score:**

A higher score would require the paper to address the generalization capabilities more convincingly, provide a fairer and more comprehensive comparison with GPT models, and offer a clearer methodological description. Additionally, the paper would benefit from a thorough analysis of trade-offs involved in specializing the model for arithmetic tasks.

**Justification For Why Not Lower Score:**

N/A

---

### Decision · Program_Chairs · 2024-01-16

Reject